# RETHINKING BACKDOOR ATTACKS ON DATASET DISTILLATION: A KERNEL METHOD PERSPECTIVE

**Ming-Yu Chung**
National Taiwan University

**Sheng-Yen Chou**
The Chinese University of Hong Kong

**Chia-Mu Yu**
National Yang Ming Chiao Tung University

**Pin-Yu Chen**
IBM Research

**Sy-Yen Kuo**
National Taiwan University

**Tsung-Yi Ho**
The Chinese University of Hong Kong

## ABSTRACT

Dataset distillation offers a potential means to enhance data efficiency in deep learning. Recent studies have shown its ability to counteract backdoor risks present in original training samples. In this study, we delve into the theoretical aspects of backdoor attacks and dataset distillation based on kernel methods. We introduce two new theory-driven trigger pattern generation methods specialized for dataset distillation. Following a comprehensive set of analyses and experiments, we show that our optimization-based trigger design framework informs effective backdoor attacks on dataset distillation. Notably, datasets poisoned by our designed trigger prove resilient against conventional backdoor attack detection and mitigation methods. Our empirical results validate that the triggers developed using our approaches are proficient at executing resilient backdoor attacks. [1]

## 1 INTRODUCTION

In recent years, deep neural networks have achieved significant success in many fields, such as natural language modeling, computer vision, medical diagnosis, etc. These successes are usually built on large-scale datasets consisting of millions or even billions of samples. Under this scale of datasets, training a model becomes troublesome because of the need for sufficiently large memory to store the datasets or the need for special infrastructure to train a model. To deal with this problem, dataset distillation (Wang et al., 2018) or dataset condensation (Zhao et al., 2021) is designed to compress the information of large datasets into a small synthetic dataset. These small datasets generated by dataset distillation, called distilled datasets, still retain a certain degree of utility. Under the same model (neural network) structure, the performance of the model trained on the distilled dataset is only slightly lower than that of the model trained on the original large-scale dataset.

However, with the development of dataset distillation techniques, the related security and privacy issues started to emerge (Liu et al., 2023a;c; Dong et al., 2022). In this paper, we focus on backdoor attacks on dataset distillation. In particular, as each distilled sample does not have a clear connection to the original samples, a straightforward stealthy backdoor attack is to poison a benign dataset first and then derive the corresponding distilled poisoned dataset. One can expect that the triggers can hardly be detected visually in the distilled poisoned dataset. However, these triggers, if not designed properly, can be diluted during dataset distillation, making backdoor attacks ineffective.

Liu et al. (2023c) empirically demonstrate the feasibility of generating a poisoned dataset surviving dataset distillation. In particular, Liu et al. (2023c) propose DOORPING as a distillation-resilient backdoor. However, DOORPING suffers from two major weaknesses. First, the resiliency and optimality of a backdoor against dataset distillation remain unclear, mainly due to the lack of a theoret-

---

[1]Code is available at `https://github.com/Mick048/KIP-based-backdoor-attack.git`.

ical foundation for the distillation resiliency. Second, DOORPING relies on a bi-level optimization and, as a consequence, consumes a significant amount of time to generate backdoor triggers.

To bridge this gap, this paper makes a step toward dataset distillation-resilient backdoors with a theoretical foundation. Our contributions can be summarized as follows:

- To the best of our knowledge, we establish the first theoretical framework to characterize backdoor effects on dataset distillation, which explains why certain backdoors survive dataset distillation.

- We propose two theory-induced backdoors, simple-trigger and relax-trigger. In particular, relax-trigger and DOORPING share the same clean test accuracy (CTA) and attack success rate (ASR). However, relax-trigger relies only on ordinary (single-level) optimization procedures and can be computationally efficient.

- We experimentally show both simple-trigger and relax-trigger signify the advanced threat vector to either completely break or weaken eight existing defenses. In particular, relax-trigger can evade all eight existing backdoor detection and cleansing methods considered in this paper.

## 2 BACKGROUND AND RELATED WORKS

**Dataset Distillation.** Dataset distillation is a technique for compressing the information of a target dataset into a small synthetic dataset. The explicit definition can be described as follows. Consider the input space $\mathcal{X} \subset \mathbb{R}^d$, the label space $\mathcal{Y} \subset \mathbb{R}^C$, and the distribution $(x, y) \sim \mathcal{D}$, where $x \in \mathcal{X}$ and $y \in \mathcal{Y}$. Suppose we are given a dataset denoted by $\mathcal{T} = \{(x_t, y_t)\}_{t=1}^{N} \sim \mathcal{D}^N$ where $x_t \in \mathcal{X}$, $y_t \in \mathcal{Y}$, and $N$ is the number of samples, and a synthetic dataset denoted as $\mathcal{S} = \{(x_s, y_s)\}_{s=1}^{N_\mathcal{S}}$ where $x_s \in \mathcal{X}$, $y_s \in \mathcal{Y}$, $N_\mathcal{S}$ is the number of samples in $\mathcal{S}$, and $N_\mathcal{S} \ll N$. The synthetic dataset $\mathcal{S}^*$ generated by a dataset distillation method can be formulated as

$$\mathcal{S}^* = \arg\min_{\mathcal{S}} \mathcal{L}(\mathcal{S}, \mathcal{T}), \tag{1}$$

where $\mathcal{L}$ is some function to measure the information loss between $\mathcal{S}$ and $\mathcal{T}$. There are several types of $\mathcal{L}$. One of the most straightforward ways to define $\mathcal{L}$ is to measure the model's performance. In this sense, the dataset distillation can be reformulated as

$$\mathcal{S}^* = \arg\min_{\mathcal{S}} \frac{1}{N} \ell(f_\mathcal{S}, \mathcal{T}) \text{ subject to } f_\mathcal{S} = \arg\min_{f \in \mathcal{H}} \frac{1}{N_\mathcal{S}} \ell(f, \mathcal{S}) + \lambda \|f\|_\mathcal{H}^2 \tag{2}$$

where the model (a classifier) is denoted as $f : \mathcal{X} \to \mathcal{Y}$, $\mathcal{H}$ is some collection of models (hypothesis class), $\ell$ is the loss function measuring the loss of model evaluated on the dataset, $\lambda \geq 0$ is the weight for the regularization term, and $\|\|_\mathcal{H}$ is some norm defined on $\mathcal{H}$. Eq. (2) forms a bi-level optimization problem. This type of dataset distillation is categorized as *performance-matching dataset distillation* in (Yu et al., 2023). For example, all of the methods from (Wang et al., 2018; Nguyen et al., 2021; Loo et al., 2022; Zhou et al., 2022; Loo et al., 2023) are performance-matching dataset distillation, while the methods from (Zhao & Bilen, 2023; Lee et al., 2022a; Wang et al., 2022; Zhao et al., 2021; Lee et al., 2022b; Liu et al., 2022; 2023b; Wang et al., 2023) belong to either parameter-preserving or distribution-preserving. In this paper, we focus only on performance-matching dataset distillation, with a particular example on kernel inducing points (KIP) from Nguyen et al. (2021).

**Reproducing Kernel Hilbert Space and KIP.** In general, the inner optimization problem in Eq. (2) does not have a closed-form solution, which not only increases the computational cost, but also increases the difficulty of analyzing this problem. To alleviate this problem, we assume our model lies in the reproducing kernel Hilbert space (RKHS) (Aronszajn, 1950; Berlinet & Thomas-Agnan, 2011; Ghojogh et al., 2021).

**Definition 1** (Kernel). *$k : \mathcal{X} \times \mathcal{X} \to \mathbb{R}$ is a kerenl if the following two points hold. (a) $\forall x, x' \in \mathcal{X}$, the kernel $k$ is symmetric; i.e., $k(x, x') = k(x', x)$. (b) $\forall n \in \mathbb{N}$, $\forall \{x_1, x_2, \ldots, x_n\}$ where each $x_i$ are sampled from $\mathcal{X}$, the kernel matrix $\boldsymbol{K}$ defined as $\boldsymbol{K}_{ij} := k(x_i, x_j)$ is postive semi-definite.*

**Definition 2** (Reproducing Kernel Hilbert Space). *Given an kernel $k : \mathcal{X} \times \mathcal{X} \to \mathbb{R}$, the collection of real-valued model $\mathcal{H}_k = \{f : \mathcal{X} \to \mathbb{R}\}$ is a reproducing kernel Hilbert space corresponding to the kernel $k$, if (a) $\mathcal{H}_k$ is a Hilbert space corresponding to the inner product $\langle \cdot, \cdot \rangle_{\mathcal{H}_k}$, (b) $\forall x \in \mathcal{X}$, $k(\cdot, x) \in \mathcal{H}_k$, (c) $\forall x \in \mathcal{X}$ and $f \in \mathcal{H}_k$, $f(x) = \langle f, k(\cdot, x) \rangle_{\mathcal{H}_k}$ (Reproducing property).*

There are several advantages to considering RKHS for solving optimization problems. One of the most beneficial properties is that there is Representer Theorem (Kimeldorf & Wahba, 1971; Ghojogh et al., 2021) induced by the reproducing property. In particular, consider the optimization problem:

$$f^* = \arg\min_{f \in \mathcal{H}_k} \frac{1}{N} \sum_{i=1}^{N} \ell(f(x_i), y_i) + \lambda \|f\|_{\mathcal{H}_k}^2, \tag{3}$$

where $f : \mathcal{X} \to \mathbb{R}$, $y_i \in \mathbb{R}$, $\lambda \geq 0$ is the weight for the regularization term. The solution of the optimization problem $f^*$ can be expressed as the linear combination of $\{k(\cdot, x_i)\}_i^N$. Furthermore, if we set $\ell(f, (x, y)) = \|f(x) - y\|_2^2$, there is a closed-form expression for $f^*$:

$$f^*(x) = k(x, \boldsymbol{X})[k(\boldsymbol{X}, \boldsymbol{X}) + N\lambda \boldsymbol{I}]^{-1} \boldsymbol{Y}, \tag{4}$$

where $k(x, \boldsymbol{X}) = [k(x, x_1), k(x, x_2), \ldots, k(x, x_N)]$, $k(\boldsymbol{X}, \boldsymbol{X})$ is an $N \times N$ matrix with $[k(\boldsymbol{X}, \boldsymbol{X})]_{ij} = k(x_i, x_j)$, and $\boldsymbol{Y} = [y_1, y_2, \ldots, y_N]^T$. Now, we return to Eq. (2). By rewriting the model $f : \mathcal{X} \to \mathcal{Y} \subset \mathbb{R}^c$ as $[f^1, f^2, \cdots, f^c]^T$, where each $f^i : \mathcal{X} \to \mathbb{R}$ is a real-valued function and $f^i$ is bounded in the RKHS $\mathcal{H}_k$, the inner optimization problem for $f_{\mathcal{S}}$ in Eq. (2) can be considered as $c$ independent optimization problems and each problem has a closed-form solution as shown in Eq. (4). Thus, the solution of the inner optimization problem can be expressed as

$$f_{\mathcal{S}}(x)^T = k(x, \boldsymbol{X}_{\mathcal{S}})[k(\boldsymbol{X}_{\mathcal{S}}, \boldsymbol{X}_{\mathcal{S}}) + N_{\mathcal{S}}\lambda \boldsymbol{I}]^{-1} \boldsymbol{Y}_{\mathcal{S}}, \tag{5}$$

where $k(x, \boldsymbol{X}_{\mathcal{S}}) = [k(x, x_{s_1}), k(x, x_{s_2}), \ldots, k(x, x_{s_{N_{\mathcal{S}}}})]$, $k(\boldsymbol{X}_{\mathcal{S}}, \boldsymbol{X}_{\mathcal{S}})$ is a $N_{\mathcal{S}} \times N_{\mathcal{S}}$ matrix with $[k(\boldsymbol{X}_{\mathcal{S}}, \boldsymbol{X}_{\mathcal{S}})]_{ij} = k(x_{s_i}, x_{s_j})$, and $\boldsymbol{Y}_{\mathcal{S}}$ is a $N_{\mathcal{S}} \times c$ matrix with $\boldsymbol{Y}_{\mathcal{S}} = [y_{s_1}, y_{s_2}, \ldots, y_{s_{N_{\mathcal{S}}}}]^T$.

Then, the dataset distillation problem can be expressed as

$$\mathcal{S}^* = \arg\min_{\mathcal{S}} \frac{1}{N} \sum_{t=1}^{N} \|f_{\mathcal{S}}(x_t) - y_t\|_2^2, \tag{6}$$

where $f_{\mathcal{S}}(x)^T = k(x, \boldsymbol{X}_{\mathcal{S}})[k(\boldsymbol{X}_{\mathcal{S}}, \boldsymbol{X}_{\mathcal{S}}) + N_{\mathcal{S}}\lambda \boldsymbol{I}]^{-1} \boldsymbol{Y}_{\mathcal{S}}$ as shown in Eq. (5). We reduce a two-level optimization problem to a one-level optimization problem using RKHS. Essentially, KIP (Nguyen et al., 2021) can be formulated as Eq. (6).

An important problem for Eq. (6) is how to construct or select a kernel $k(\cdot, \cdot)$. Nevertheless, we do not discuss this problem in this paper. We directly consider the neural tangent kernel (NTK) (Jacot et al., 2018; He et al., 2020; Lee et al., 2019) induced by a three-layer neural network as the kernel $k(\cdot, \cdot)$ to do the experiment in Section 4.

**Backdoor Attack.** Backdoor attack introduces some malicious behavior into the model without degrading the model's performance on the original task by poisoning the dataset (Gu et al., 2019; Chen et al., 2017; Liu et al., 2018b; Turner et al., 2019; Nguyen & Tran, 2020; Barni et al., 2019; Li et al., 2021c; Nguyen & Tran, 2021; Liu et al., 2020; Tang et al., 2021; Qi et al., 2022; Souri et al., 2022). To be more specific, consider the following scenario. Suppose there are two types of distributions, $(x_a, y_a) \sim \mathcal{D}_A$ and $(x_b, y_b) \sim \mathcal{D}_B$. $\mathcal{D}_A$ corresponds to the original normal behavior, while $\mathcal{D}_B$ corresponds to the malicious behavior. The goal of the backdoor attack is to construct a poisoned dataset such that the model trained on it learns well for both the original normal distribution $\mathcal{D}_A$ and the malicious distribution $\mathcal{D}_B$. In other words, an attacker wants to construct a dataset $\tilde{D}$ such that the model trained on $\tilde{D}$, denoted $f_{\tilde{D}}$, has sufficiently low risk $\mathbb{E}_{(x_a, y_a) \sim \mathcal{D}_A} \ell(f_{\tilde{D}}, (x_a, y_a))$ and $\mathbb{E}_{(x_b, y_b) \sim \mathcal{D}_B} \ell(f_{\tilde{D}}, (x_b, y_b))$ at the same time.

One approach to constructing such a dataset $\tilde{D}$ is to directly mix the *benign dataset* $D_A \sim \mathcal{D}_A^{N_A}$ and the *trigger dataset* $D_B \sim \mathcal{D}_B^{N_B}$. An attacker usually wants to make the attack stealthy, and so it sets $N_B \ll N_A$. We define $\mathcal{D}_B$ according to the original normal behavior $\mathcal{D}_A$, the trigger $T \in \mathbb{R}^d$, and the trigger label $y_T \in \mathcal{Y}$:

$$(x_b, y_b) := ((1 - m) \odot x_a + m \odot T, y_T), \tag{7}$$

where $x_a \sim \mathcal{D}_A$, $m \in \mathbb{R}^d$ is the real-valued mask, and $\odot$ is the Hadamard product.

## 3 PROPOSED METHODS AND THEORETICAL ANALYSIS

In this paper, we aim to use dataset distillation (KIP as a representative) to perform the backdoor attack. In the simplest form of KIP-based backdoor attacks (as shown in Algorithm 1 of the Appendix), we first construct the *poisoned dataset* $\tilde{D} = D_A \cup D_B$ from $\mathcal{D}_A^{N_A}$ and $\mathcal{D}_B^{N_B}$. Then, we perform KIP on $\tilde{D}$ and compress the information in $\tilde{D}$ into the *distilled poisoned dataset* $\mathcal{S}^* = \{(x_s, y_s)\}_{s=1}^{N_{\mathcal{S}}}$, where $N_{\mathcal{S}} \ll N_A + N_B$. Namely, we solve the following optimization problem

$$\mathcal{S}^* = \arg\min_{\mathcal{S}} \frac{1}{N_A + N_B} \sum_{(x,y) \in \tilde{D}} \|f_{\mathcal{S}}(x) - y\|_2^2, \tag{8}$$

where $f_{\mathcal{S}}(x)^T = k(x, \boldsymbol{X}_{\mathcal{S}})[k(\boldsymbol{X}_{\mathcal{S}}, \boldsymbol{X}_{\mathcal{S}}) + N_{\mathcal{S}}\lambda \boldsymbol{I}]^{-1}\boldsymbol{Y}_{\mathcal{S}}$. Essentially, the above KIP-based backdoor attack is the same as Naive attack in (Liu et al., 2023c) except that the other distillation, instead of KIP, is used in Naive attack. The experimental results in (Liu et al., 2023c) show that ASR grows but CTA drops significantly when the trigger size increases. Liu et al. (2023c) claims a trade-off between CTA and the trigger size. Nonetheless, we find that our KIP-based backdoor attack does not have such a trade-off. This motivates us to develop a theoretical framework for backdoor attacks on dataset distillation.

Below, we introduce the theoretical framework in Section 3.1, followed by two theory-induced backdoor attacks, simple-trigger and relax-trigger in Section 3.2 and Section 3.3, respectively.

### 3.1 THEORETICAL FRAMEWORK

We first introduce the structure of our analysis, which divides the risk of KIP-based backdoor attacks into three parts: projection loss, conflict loss, and generalization gap. Then, we provide an upper bound for each part of the risk.

**Structure of Analysis.** Recall that the goal of a KIP-based backdoor attack is to construct the synthetic dataset $\mathcal{S}^*$ such that the risk $\mathbb{E}_{(x,y)\sim\mathcal{D}}\ell(f_{\mathcal{S}^*}, (x, y))$ is sufficiently low, where $\mathcal{D}$ is the normal distribution $\mathcal{D}_A$ or the malicious distribution $\mathcal{D}_B$. The classical framework for analyzing this problem is to divide the risk into two parts, the empirical risk and generalization gap. Namely,

$$\mathbb{E}_{(x,y)\sim\mathcal{D}}\,\ell(f_{\mathcal{S}^*}, (x, y)) = \underbrace{\mathbb{E}_{(x,y)\sim D}\,\ell(f_{\mathcal{S}^*}, (x, y))}_{\text{Empirical risk}}$$
$$+ \underbrace{[\mathbb{E}_{(x,y)\sim\mathcal{D}}\,\ell(f_{\mathcal{S}^*}, (x, y)) - \mathbb{E}_{(x,y)\sim D}\,\ell(f_{\mathcal{S}^*}, (x, y))]}_{\text{Generalization gap}} \tag{9}$$

where $D = \{(x_i, y_i)\}_{i=1}^N$ is the dataset sampled from the distribution $\mathcal{D}^N$ and $N$ is the number of samples of $D$. Here, we consider that $D$ is $D_A \sim \mathcal{D}^{N_A}$ or $D_B \sim \mathcal{D}^{N_B}$. In our framework, we continue to divide the empirical risk into two parts as

$$\mathbb{E}_{(x,y)\sim D}\,\ell(f_{\mathcal{S}^*}, (x, y)) \leq \frac{N_A + N_B}{N}[\underbrace{\min_{\mathcal{S}}\mathbb{E}_{(x,y)\sim\tilde{D}}\ell(f_{\mathcal{S}}, (x, f_{\tilde{D}}(x)))}_{\text{Projection Loss}} + \underbrace{\mathbb{E}_{(x,y)\sim\tilde{D}}\ell(f_{\tilde{D}}, (x, y))}_{\text{Conflict Loss}}]$$
$$\tag{10}$$

where $\tilde{D} = D_A \cup D_B$, $f_{\tilde{D}}$ is the model trained on $\tilde{D}$ with the weight of the regularization term $\lambda \geq 0$ and $f_{\mathcal{S}}$ is the model trained on $\mathcal{S}$ with the weight of the regularization term $\lambda_{\mathcal{S}} \geq 0$. Intuitively, given a dataset $\tilde{D}$ constructed from $\mathcal{D}_A^{N_A}$ and $\mathcal{D}_B^{N_B}$, $f_{\tilde{D}}$ is regarded as the best model derived from the information of $\tilde{D}$. The conflict loss reflects the internal information conflict between the information about $\mathcal{D}_A$ in $\tilde{D}$ and the information about $\mathcal{D}_B$ in $\tilde{D}$. For example, we consider a dog/cat picture classification problem. In the dataset $D_A$, we label the dog pictures with $0$ and label the cat pictures with $1$. However, in the dataset $D_B$, we label the dog pictures with $1$ and label the cat pictures with $0$. It is clear that the model trained on $\tilde{D}$ must perform terribly on the dataset either $D_A$ or $D_B$. In this case, the information between $D_A$ and $D_B$ have strong conflict and the conflict loss would be large. On the other hand, projection loss reflects the loss of information caused by projecting $f_{\tilde{D}}$ into $\{f_{\mathcal{S}}|\mathcal{S} = \{(x_i, y_i) \in \mathcal{X} \times \mathcal{Y}\}_{i=1}^{N_{\mathcal{S}}}\}$. We can also consider the projection loss as the increase in

information induced by compressing the information of $\tilde{D}$ into the synthetic dataset $\mathcal{S}$. Take writing an abstract for example. If we want to write a 100 words abstract to describe a 10000 words article, the abstract may suffer some lack of semantics to some degree. Such a phenomena also happens for dataset distillation. When the information of a large dataset is complex enough, the information loss for dataset distillation will be significant; When the information of a large dataset is very simple, it is possible that there is only very limited information loss. We introduce the projection loss defined above to measure this phenomenon. More details can be found below.

**Conflict Loss.** In a KIP-based backdoor attack, the dataset $\tilde{D}$ is defined as $\tilde{D} = D_A \cup D_B$, where $D_A \sim \mathcal{D}_A^{N_A}$ and $D_B \sim \mathcal{D}_B^{N_B}$. By Eq. (5) we know that the model trained on $\tilde{D}$ with the weight of the regularization term $\lambda \geq 0$ has a closed-form solution if we constrain the model in the RKHS $\mathcal{H}_k^c$ and suppose that $\ell(f, (x, y)) := \|f(x) - y\|_2^2$:

$$f_{\tilde{D}}(x)^T = k(x, \boldsymbol{X}_{AB})[k(\boldsymbol{X}_{AB}, \boldsymbol{X}_{AB}) + (N_A + N_B)\lambda \boldsymbol{I}]^{-1}\boldsymbol{Y}_{AB}, \tag{11}$$

where $(N_A + N_B) \times d$ matrix $\boldsymbol{X}_{AB}$ is the matrix corresponding to the features of $\tilde{D}$, $(N_A + N_B) \times c$ matrix $\boldsymbol{Y}_{AB}$ is the matrix corresponding to the labels of $\tilde{D}$, $k(x, \boldsymbol{X}_{AB})$ is a $1 \times (N_A + N_B)$ matrix, $k(\boldsymbol{X}_{AB}, \boldsymbol{X}_{AB})$ is a $(N_A + N_B) \times (N_A + N_B)$ matrix with $[k(\boldsymbol{X}_{AB}, \boldsymbol{X}_{AB})]_{ij} = k(x_i, x_j)$, and $\boldsymbol{Y}_{AB}$ is a $(N_A + N_B) \times c$ matrix with $\boldsymbol{Y}_{AB} = [y_1, y_2, \ldots, y_{(N_A+N_B)}]^T$. Hence, we can express the conflict loss $\mathcal{L}_{\text{conflict}}$ as

$$\mathcal{L}_{\text{conflict}} = \frac{1}{N_A + N_B}\|\boldsymbol{Y}_{AB} - k(\boldsymbol{X}_{AB}, \boldsymbol{X}_{AB})[k(\boldsymbol{X}_{AB}, \boldsymbol{X}_{AB}) + (N_A + N_B)\lambda \boldsymbol{I}]^{-1}\boldsymbol{Y}_{AB}\|_2^2. \tag{12}$$

We can obtain the upper bound of $\mathcal{L}_{\text{conflict}}$ as Theorem 1.

**Theorem 1** (Upper bound of conflict loss). *The conflict loss $\mathcal{L}_{conflict}$ can be bounded as*

$$\mathcal{L}_{conflict} \leq \frac{1}{N_A + N_B}Tr(\boldsymbol{I} - k(\boldsymbol{X}_{AB}, \boldsymbol{X}_{AB})[k(\boldsymbol{X}_{AB}, \boldsymbol{X}_{AB}) + (N_A + N_B)\lambda \boldsymbol{I}]^{-1})^2\|\boldsymbol{Y}_{AB}\|_2^2 \tag{13}$$

*where Tr is the trace operator, $k(\boldsymbol{X}_{AB}, \boldsymbol{X}_{AB})$ is a $(N_A + N_B) \times (N_A + N_B)$ matrix, and $\boldsymbol{Y}_{AB}$ is a $(N_A + N_B) \times c$ matrix.*

The proof of Theorem 1 can be found in Appendix A.3. From Theorem 1, we know that the conflict loss can be characterized by $Tr(\boldsymbol{I} - k(\boldsymbol{X}_{AB}, \boldsymbol{X}_{AB})[k(\boldsymbol{X}_{AB}, \boldsymbol{X}_{AB}) + (N_A + N_B)\lambda \boldsymbol{I}]^{-1})$. However, in the latter sections, we do not utilize $Tr(\boldsymbol{I} - k(\boldsymbol{X}_{AB}, \boldsymbol{X}_{AB})[k(\boldsymbol{X}_{AB}, \boldsymbol{X}_{AB}) + (N_A + N_B)\lambda \boldsymbol{I}]^{-1})$ to construct trigger pattern generalization algorithm; instead, we use Eq. (12) directly. It is because Eq. (12) can be computed more precisely although $Tr(\boldsymbol{I} - k(\boldsymbol{X}_{AB}, \boldsymbol{X}_{AB})[k(\boldsymbol{X}_{AB}, \boldsymbol{X}_{AB}) + (N_A + N_B)\lambda \boldsymbol{I}]^{-1})$ and Eq. (12) have similar computational cost.

**Projection Loss.** To derive the upper bound of the projection loss, we first derive Lemma 1.

**Lemma 1** (Projection lemma). *Given a synthetic dataset $\mathcal{S} = \{(x_s, y_s)\}_{s=1}^{N_\mathcal{S}}$, and a dataset $\tilde{D} = \{(x_i, y_i)\}_{i=1}^{N_A+N_B}$ where $(N_A + N_B)$ is the number of the samples of $\tilde{D}$. Suppose the kernel matrix $k(\boldsymbol{X}_\mathcal{S}, \boldsymbol{X}_\mathcal{S})$ is invertible, then we have*

$$k(\cdot, x_i) = \underbrace{k(\cdot, \boldsymbol{X}_\mathcal{S})k(\boldsymbol{X}_\mathcal{S}, \boldsymbol{X}_\mathcal{S})^{-1}k(\boldsymbol{X}_\mathcal{S}, x_i)}_{\in \mathcal{H}_\mathcal{S}} + \underbrace{[k(\cdot, x_i) - k(\cdot, \boldsymbol{X}_\mathcal{S})k(\boldsymbol{X}_\mathcal{S}, \boldsymbol{X}_\mathcal{S})^{-1}k(\boldsymbol{X}_\mathcal{S}, x_i)]}_{\in \mathcal{H}_\mathcal{S}^\perp}, \quad \forall(x_i, y_i) \in \tilde{D} \tag{14}$$

*where $\mathcal{H}_\mathcal{S} := span(\{k(\cdot, x_s) \in \mathcal{H}_k | (x_s, y_s) \in \mathcal{S}\})$ and $\mathcal{H}_\mathcal{S}^\perp$ is the collection of functions orthogonal to $\mathcal{H}_\mathcal{S}$ corresponding to the inner product $\langle \cdot, \cdot \rangle_{\mathcal{H}_k}$. Thus, $k(\cdot, \boldsymbol{X}_\mathcal{S})k(\boldsymbol{X}_\mathcal{S}, \boldsymbol{X}_\mathcal{S})^{-1}k(\boldsymbol{X}_\mathcal{S}, x_i)$ is the solution of the optimization problem:*

$$\underset{f \in \mathcal{H}_S}{\arg\min} \sum_{(x_s, y_s) \in \mathcal{S}} \|f(x_s) - k(x_s, x_i)\|_2^2. \tag{15}$$

The proof of Lemma 1 can be found in Appendix A.2. Now, we turn to the scenario of the KIP-based backdoor attack. Given a mixed dataset $\tilde{D} = D_A \cup D_B$ where $D_A \sim \mathcal{D}_A^{N_A}$ and $D_B \sim \mathcal{D}_B^{N_B}$. We also constrained models in the RKHS $\mathcal{H}_k^c$ and suppose $\ell(f, (x, y)) := \|f(x) - y\|_2^2$. With the help of Lemma 1, we can obtain the following theorem:

**Theorem 2** (Upper bound of projection loss). *Suppose the kernel matrix of the synthetic dataset $k(\boldsymbol{X}_{\mathcal{S}}, \boldsymbol{X}_{\mathcal{S}})$ is invertible, $f_{\mathcal{S}}$ is the model trained on the synthetic dataset $\mathcal{S}$ with the regularization term $\lambda_{\mathcal{S}}$, where the projection loss $\mathcal{L}_{project} = \min_{\mathcal{S}} \mathbb{E}_{(x,y)\sim\tilde{D}} \ell(f_{\mathcal{S}}, (x, f_{\tilde{D}}(x)))$ can be bounded as*

$$\mathcal{L}_{project} \leq \sum_{(x_i, y_i) \in \tilde{D}} \min_{\boldsymbol{X}_{\mathcal{S}}} \sum_{j=1}^{c} \frac{|\alpha_{i,j}|^2}{N_A + N_B} \|k(\boldsymbol{X}_{AB}, x_i) - k(\boldsymbol{X}_{AB}, \boldsymbol{X}_{\mathcal{S}})k(\boldsymbol{X}_{\mathcal{S}}, \boldsymbol{X}_{\mathcal{S}})^{-1}k(\boldsymbol{X}_{\mathcal{S}}, x_i)\|_2^2. \quad (16)$$

*where $\alpha_{i,j} := [[k(\boldsymbol{X}_{AB}, \boldsymbol{X}_{AB}) + (N_A + N_B)\lambda \boldsymbol{I}]^{-1}\boldsymbol{Y}_{AB}]_{i,j}$, which is the weight of $k(\cdot, x_i)$ corresponding to $f_{\tilde{D}}^j$, $\boldsymbol{X}_{AB}$ is the $(N_A + N_B) \times d$ matrix corresponding to the features of $\tilde{D}$, $\boldsymbol{X}_{\mathcal{S}}$ is the $N_{\mathcal{S}} \times d$ matrix corresponding to the features of $\mathcal{S}$, $\boldsymbol{Y}_{AB}$ is the $(N_A + N_B) \times c$ matrix corresponding to the labels of $\tilde{D}$, $\boldsymbol{Y}_{\mathcal{S}}$ is the $N_{\mathcal{S}} \times c$ matrix corresponding to the labels of $\mathcal{S}$.*

The proof of Theorem 2 can be found in Appendix A.4. In Theorem 2, we first characterize the natural information loss when compressing the information of $\tilde{D}$ into an arbitrary dataset $\mathcal{S}$, and then bound the information loss for the synthetic dataset $\mathcal{S}^*$ generated by dataset compression by taking the minimum. This formulation gives some insight into the construction of our trigger generation algorithm, which is discussed in the later section.

**Generalization Gap.** Finally, for the generalization gap, we follow the existing theoretical results (Theorem 3.3 in (Mohri et al., 2012)), but modify them a bit. Let $\mathcal{G} = \{g : (x, y) \mapsto \|f(x) - y\|_2^2 | f \in \mathcal{H}_k^c\}$. Assume that the distribution $\mathcal{D}$ is distributed in a bounded region, and that $\mathcal{G} \subset C^1$ and the norm of the gradient of $g \in \mathcal{G}$ have a common non-trivial upper bound. Namely, $\|(x, y) - (x', y')\|_2 \leq \Gamma_{\mathcal{D}}$ for any sample which is picked from $\mathcal{D}$ and $\|\nabla g\|_2 \leq L_{\mathcal{D}}$. Then we can obtain Theorem 3.

**Theorem 3** (Upper bound of generalization gap). *Given a $N$-sample dataset $D$, sampled from the distribution $\mathcal{D}$, the following generalization gap holds for all $g \in \mathcal{G}$ with probability at least $1 - \delta$:*

$$\mathbb{E}_{(x,y)\sim\mathcal{D}}[g((x, y))] - \sum_{(x_i, y_i) \in D} \frac{g((x_i, y_i))}{N} \leq 2\hat{\mathfrak{R}}_D(\mathcal{G}) + 3L_{\mathcal{D}}\Gamma_{\mathcal{D}}\sqrt{\frac{\log \frac{2}{\delta}}{2N}}, \quad (17)$$

*where $\boldsymbol{X}$ is the matrix of the features of $D$ and $\hat{\mathfrak{R}}_D(\mathcal{G})$ is the empirical Rademacher's complexity.*

The proof of Theorem 3 can be found in Appendix A.5. We know from Theorem 3 that the upper bound of the generalization gap is characterized by two factors, $\hat{\mathfrak{R}}_D(\mathcal{G})$ and $\Gamma_{\mathcal{D}}$. The lower $\hat{\mathfrak{R}}_D(\mathcal{G})$ and $\Gamma_{\mathcal{D}}$ imply the lower generalization gap. We usually assume $k(x, x) \leq r^2$ and $\sqrt{\langle f, f \rangle_{\mathcal{H}_k}} \leq \Lambda$ (as in Theorem 6.12 of (Mohri et al., 2012)). Under this setting, we can ignore $\hat{\mathfrak{R}}_D(\mathcal{G})$ for the upper bound of the generalization gap and only focus on $\Gamma_{\mathcal{D}}$. ASR relates to the risk for $\mathcal{D}_B$ and hence corresponds to the generalization gap evaluated on $\mathcal{D}_B$. This theoretical consequence can be used to explain the phenomenon that ASR of the backdoor attack increases as we enlarge the trigger size.

## 3.2 THEORY-INDUCED BACKDOOR: SIMPLE-TRIGGER

Consider $\mathcal{D}$ in Theorem 3 as $\mathcal{D}_B$ and the corresponding dataset $D$ as $D_B$. Conventionally, a cell of the mask $m$ in Eq. (7) is 1 it corresponds to a trigger, and is 0 otherwise. Recall that the definition of $\mathcal{D}_B$ in Eq. (7), it is clear that the $\Gamma_{\mathcal{D}_B}$ will monotonely decrease from $\Gamma_{\mathcal{D}_A}$ to 0 as we enlarge the trigger size. If we enlarge the trigger size, the $\Gamma_{\mathcal{D}_B}$ drops to zero, which implies that the corresponding generalization gap will be considerably small. Thus, the success of the large trigger pattern can be attributed to its relatively small generalization gap.

So, given an image of size $m \times n$ ($m \leq n$), simple-trigger generates a trigger of size $m \times n$. The default pattern for the trigger generated by simple-trigger is whole-white. In fact, since the generalization gap is irrelevant to the trigger pattern, we do not impost any pattern restrictions.

## 3.3 THEORY-INDUCED BACKDOOR: RELAX-TRIGGER

In simple-trigger, we optimize the trigger through only the generalization gap. However, we know that ASR can be determined by conflict loss, projection loss, and generalization gap because of

Theorems $1\sim3$ (i.e., all are related to $\mathcal{D}_B$). On the other hand, CTA is related to conflict loss and projection loss, because the generalization gap is irrelevant to CTA. That is, Eq. (17) evaluated on $\mathcal{D}_A$ is a constant as we modify the trigger. As a result, the lower conflict loss, projection loss, and generalization gap imply a backdoor attack with greater ASR and CTA. Therefore, relax-trigger aims to construct a trigger whose corresponding $\mathcal{D}_B$ make Eq. (12), Eq. (16), and $\Gamma_{\mathcal{D}_B}$ sufficiently low. The computation procedures of relax-trigger can be found in Algorithm 2 of Appendix A.7.

Suppose $D_A$, $N_A$ and $N_B$ are fixed. To reduce the bound in Eq. (12), one considers $D_B$ as a function depending on the trigger $T$ and then uses the optimizer to find the optimal trigger $T^*$. In this sense, we solve the following optimization problem

$$\arg\min_T \|\boldsymbol{Y}_{AB} - k(\boldsymbol{X}_{AB}, \boldsymbol{X}_{AB})[k(\boldsymbol{X}_{AB}, \boldsymbol{X}_{AB}) + (N_A + N_B)\lambda\boldsymbol{I}]^{-1}\boldsymbol{Y}_{AB}\|_2^2. \tag{18}$$

On the other hand, a low $\Gamma_{\mathcal{D}_B}$ can be realized by enlarging the trigger as mentioned in Section 3.2.

Finally, to make Eq. (16) sufficiently low, we consider $\mathcal{D}_B$ as a function of the trigger $T$, and then directly optimize

$$\arg\min_T \left\{ \sum_{(x_i, y_i) \in \tilde{D}} \min_{\boldsymbol{X}_{\mathcal{S}}} \sum_{j=1}^c |\alpha_{i,j}|^2 \|k(\boldsymbol{X}_{AB}, x_i) - k(\boldsymbol{X}_{AB}, \boldsymbol{X}_{\mathcal{S}})k(\boldsymbol{X}_{\mathcal{S}}, \boldsymbol{X}_{\mathcal{S}})^{-1}k(\boldsymbol{X}_{\mathcal{S}}, x_i)\|_2^2 \right\}. \tag{19}$$

However, Eq. (19) is a bi-level optimization problem that is difficult to solve. Instead, we set the synthetic dataset $\mathcal{S}$ in Eq. (19) to $\mathcal{S}_A$, which is the distilled dataset from $D_A$. Then, the two-level optimization problem can be converted into a one-level optimization problem below.

$$\arg\min_T \left\{ \sum_{(x_i, y_i) \in \tilde{D}} \sum_{j=1}^c |\alpha_{i,j}|^2 \|k(\boldsymbol{X}_{AB}, x_i) - k(\boldsymbol{X}_{AB}, \boldsymbol{X}_{\mathcal{S}_A})k(\boldsymbol{X}_{\mathcal{S}_A}, \boldsymbol{X}_{\mathcal{S}_A})^{-1}k(\boldsymbol{X}_{\mathcal{S}_A}, x_i)\|_2^2 \right\}. \tag{20}$$

Eq. (20) can be easily solved by directly applying optimizers like Adam (P. Kingma & Ba, 2015). Eq. (20) aims to find a trigger $T$ such that $\tilde{D}$ generated from $D_A$ and $D_B$ will be compressed into the neighborhood of $\mathcal{S}_A \subset (\mathcal{X} \times \mathcal{Y})^{N_{\mathcal{S}}}$, which guarantees that CTA of the model trained on the distilled $\tilde{D}$ is similar to CTA of the model trained on the distilled $D_A$. Overall, relax-trigger solves the following optimization,

$$\arg\min_T \{ \sum_{(x_i, y_i) \in \tilde{D}} \sum_{j=1}^c |\alpha_{i,j}|^2 \|k(\boldsymbol{X}_{AB}, x_i) - k(\boldsymbol{X}_{AB}, \boldsymbol{X}_{\mathcal{S}_A})k(\boldsymbol{X}_{\mathcal{S}_A}, \boldsymbol{X}_{\mathcal{S}_A})^{-1}k(\boldsymbol{X}_{\mathcal{S}_A}, x_i)\|_2^2$$
$$+ \rho\|\boldsymbol{Y}_{AB} - k(\boldsymbol{X}_{AB}, \boldsymbol{X}_{AB})[k(\boldsymbol{X}_{AB}, \boldsymbol{X}_{AB}) + (N_A + N_B)\lambda\boldsymbol{I}]^{-1}\boldsymbol{Y}_{AB}\|_2^2 \}, \tag{21}$$

where $\rho > 0$ is the penalty parameter, $m$ is the previously chosen mask, the malicious dataset is defined as $D_B = \{(x_b, y_b) = ((1-m) \odot x_a + m \odot T, y_T)|(x_a, y_a) \in D_A\}$. We particularly note that Eq. (19) is converted into Eq. (20) because we use $\mathcal{S}_A$ to replace the minimization over $\mathcal{S}$.

relax-trigger is different from DOORPING in (Liu et al., 2023c). DOORPING generates the trigger during the process of sample compression. In other words, DOORPING is induced by solving a bi-level optimization problem. However, relax-trigger is induced by a one-level optimization problem (Eq. (21)). The design rationale of relax-trigger is different from DOORPING. DOORPING aims to find the globally best trigger but consumes a significant amount of computation time. On the other hand, through our theoretical framework, relax-trigger aims to find the trigger that reliably compresses the corresponding $\tilde{D}$ into the neighborhood of our $\mathcal{S}_A$ with the benefit of time efficiency.

## 4 EVALUATION

### 4.1 EXPERIMENTAL SETTING

**Dataset.** Two datasets are chosen for measuring the backdoor performance.

- **CIFAR-10** is a 10-class dataset with 6000 $32 \times 32$ color images per class. CIFAR-10 is split into 50000 training images and 10000 testing images.
- **GTSRB** contains 43 classes of traffic signs with 39270 images, which are split into 26640 training images and 12630 testing images. We resize all images to $32 \times 32$ color images.

**Dataset Distillation and Backdoor Attack.** We use KIP (Nguyen et al., 2021) to implement backdoor attacks with the neural tangent kernel (NTK) induced by a 3-layer neural network, which has the same structure in the Colab notebook of (Nguyen et al., 2021). We also set the optimizer to Adam (P. Kingma & Ba, 2015), the learning rate to 0.01, and the batch size to $10 \times$ number of class for each dataset. We run KIP with 1000 training steps to generate a distilled dataset. We perform 3 independent runs for each KIP-based backdoor attack to examine the performance.

**Evaluation Metrics.** We consider two metrics, clean test accuracy (CTA) and attack success rate (ASR). Consider $\mathcal{S}$ as a distilled dataset from the KIP-based backdoor attack. CTA is defined as the test accuracy of the model trained on $\mathcal{S}$ and evaluated on the normal (clean) test dataset, while ASR is defined as the test accuracy of the model trained on $\mathcal{S}$ and evaluated on the trigger test dataset.

**Defense for Backdoor Attack.** In this paper we consider eight existing defenses, SCAn (Tang et al., 2021), AC (Chen et al., 2018), SS (Tran et al., 2018), Strip (modified as a poison cleaner) (Gao et al., 2019), ABL (Li et al., 2021a), NAD (Li et al., 2021b), STRIP (backdoor input filter) (Gao et al., 2019), FP (Liu et al., 2018a), to investigate the ability to defend against KIP-based backdoor attack. The implementation of the above defenses is from the backdoor-toolbox[2].

## 4.2 EXPERIMENTAL RESULTS

**Performance of simple-trigger.** We performed a series of experiments to demonstrate the effectiveness of simple-trigger. In our setting, $N_\mathcal{S}$ is set to $10 \times$ number of classes and $50 \times$ number of classes for each dataset. We also configured the trigger as $2 \times 2$, $4 \times 4$, $8 \times 8$, $16 \times 16$, $32 \times 32$ white square patterns. The corresponding results are shown in Table 1. The experiment results suggest that CTA and ASR of simple-trigger increase as we enlarge the trigger size, which is consistent with our theoretical analysis (Theorem 3). One can see that for the $32 \times 32$ white square trigger, ASR can achieve $100\%$ without sacrificing CTA.

| Data. (Size)\Trig. | None | $2 \times 2$ | | $4 \times 4$ | | $8 \times 8$ | | $16 \times 16$ | | $32 \times 32$ | |
|---|---|---|---|---|---|---|---|---|---|---|---|
| | CTA (%) | CTA (%) | ASR (%) | CTA (%) | ASR (%) | CTA (%) | ASR (%) | CTA (%) | ASR (%) | CTA (%) | ASR (%) |
| CIFAR-10 (100) | 42.55 (0.13) | 41.78 (0.22) | 65.73 (0.80) | 41.53 (0.31) | 90.59 (0.22) | 41.46 (0.32) | 98.29 (0.18) | 41.55 (0.43) | 99.94 (0.05) | 41.70 (0.25) | 100.00 (0.00) |
| CIFAR-10 (500) | 44.52 (0.23) | 43.89 (0.13) | 82.36 (0.39) | 43.85 (0.23) | 92.89 (0.16) | 43.60 (0.23) | 98.19 (0.16) | 43.70 (0.40) | 99.88 (0.07) | 43.66 (0.40) | 100.00 (0.00) |
| GTSRB (430) | 69.27 (0.19) | 67.06 (0.74) | 74.14 (0.50) | 67.01 (0.69) | 81.46 (0.37) | 66.98 (0.64) | 89.63 (0.58) | 67.10 (0.63) | 98.43 (0.13) | 67.56 (0.60) | 100.00 (0.00) |
| GTSRB (2150) | 72.07 (0.20) | 70.87 (0.27) | 76.79 (1.08) | 70.90 (0.25) | 81.93 (0.62) | 70.92 (0.31) | 90.48 (0.74) | 70.98 (0.22) | 98.89 (0.17) | 71.27 (0.24) | 100.00 (0.00) |

Table 1: Performance of simple-trigger on CIFAR-10 and GTSRB (mean and standard deviation).

**Performance of relax-trigger.** Here, we relax the setting of the mask $m$; i.e., each component of $m$ is defined to be $0.3$, instead of $1$. This can be regarded as an increase in the trigger's transparency (the level of invisibility) for mixing an image and the trigger. Recall the definition of $\mathcal{D}_B$ in (Eq. 7). From theory point of view, under such a mask $m$, $\Gamma_{\mathcal{D}_B}$ will drop to $0.3 * \Gamma_{\mathcal{D}_A} > 0$, as we enlarge the trigger. Hence, we cannot reduce the generalization gap considerably as in the experiments of simple-trigger. It turns out that to derive better CTA and ASR, we resort to consider relax-trigger.

The result is presented in Table 2. We compare the performance (CTA and ASR) between simple-trigger ($32 \times 32$ white square), DOORPING and relax-trigger. For CIFAR-10, relax-trigger increases the ASR about $24\%$ from simple-trigger without losing CTA. For GTSRB, relax-trigger not only increases the ASR about $30\%$, but also slightly increases the CTA. On the other hand, relax-trigger possesses higher CTA and ASR compared to DOORPING. These results confirm the effectiveness of relax-trigger. The trigger patterns of relax-trigger are visualized in Figure 1.

| Dataset | Size\Trig. | simple-trigger (baseline) | | relax-trigger | | DOORPING | |
|---|---|---|---|---|---|---|---|
| | | CTA (%) | ASR (%) | CTA (%) | ASR (%) | CTA (%) | ASR (%) |
| CIFAR-10 | 100 | 41.40 (0.06) | 75.92 (1.19) | 41.66 (0.74) | 100.00 (0.00) | 36.35 (0.42) | 80.00 (40.00) |
| CIFAR-10 | 500 | 42.98 (0.13) | 75.79 (0.58) | 43.64 (0.40) | 100.00 (0.00) | | |
| GTSRB | 430 | 67.02 (0.07) | 62.74 (0.23) | 68.73 (0.67) | 95.26 (0.54) | 68.03 (0.92) | 90.00 (30.00) |
| GTSRB | 2150 | 70.28 (0.07) | 62.65 (1.12) | 71.54 (0.33) | 95.08 (0.33) | | |

Table 2: Performance of relax-trigger on CIFAR-10 and GTSRB (mean and standard deviation).

**Off-the-shelf Backdoor Defenses.** We examine whether simple-trigger and relax-trigger can survive backdoor detection and cleansing. Here, we utilize backdoor-toolbox and retrain the distilled dataset on ResNet (default setting in backdoor-toolbox) to compute CTA and ASR. In our experimental results, the term "None" denotes no defense.

---

[2] Available at `https://github.com/vtu81/backdoor-toolbox`.

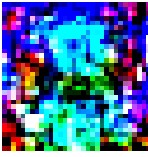 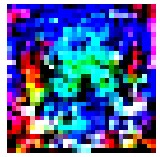 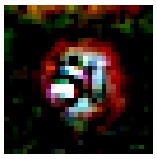 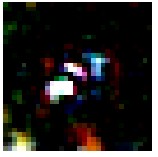

(a) CIFAR10 (size 100)   (b) CIFAR10 (size 500)   (c) GTSRB (size 430)   (d) GTSRB (size 2150)

Figure 1: Triggers generated by relax-trigger for GTSRB and CIFAR.

| Trig.\Def. | None | | SCAn | | AC | | SS | | Strip | |
|---|---|---|---|---|---|---|---|---|---|---|
| | CTA (%) | ASR (%) | CTA (%) | ASR (%) | CTA (%) | ASR (%) | CTA (%) | ASR (%) | CTA (%) | ASR (%) |
| 2 × 2 | 23.18 (1.24) | 13.98 (8.36) | 24.39 (2.23) | 18.17 (2.74) | 23.84 (1.12) | 10.57 (3.52) | 22.60 (1.62) | 11.64 (1.42) | 25.08 (0.48) | 12.79 (5.25) |
| 4 × 4 | 23.18 (1.40) | 25.26 (9.67) | 24.67 (0.97) | 15.73 (3.69) % | 24.37 (1.29) | 14.00 (5.84) | 21.98 (3.30) | 10.81 (2.44) | 24.28 (0.39) | 17.68 (5.47) |
| 8 × 8 | 25.69 (1.06) | 13.35 (5.38) | 26.40 (0.11) | 14.08 (3.72) | 23.24 (1.96) | 9.19 (5.53) | 21.49 (1.77) | 7.10 (4.61) | 25.13 (0.74) | 12.09 (5.52) |
| 16 × 16 | 25.90 (5.76) | 81.29 (2.96) | 26.39 (2.96) | 49.66 (9.66) | 25.85 (1.57) | 55.26 (10.94) | 24.03 (1.66) | 40.03 (27.29) | 26.22 (0.75) | 41.36 (40.18) |
| 32 × 32 | 28.95 (1.56) | 100.00 (0.00) | 28.28 (1.45) | 66.67 (47.14) | 25.35 (2.05) | 66.67 (47.14) | 22.21 (1.02) | 66.67 (47.14) | 25.68 (2.04) | 0.00 (0.00) |

| Trig.\Def. | | | ABL | | NAD | | STRIP | | FP | |
|---|---|---|---|---|---|---|---|---|---|---|
| | | | CTA (%) | ASR (%) | CTA (%) | ASR (%) | CTA (%) | ASR (%) | CTA (%) | ASR (%) |
| 2 × 2 | | | 13.31 (2.43) | 1.38 (1.14) | 31.74 (1.90) | 5.45 (0.78) | 20.91 (1.07) | 12.63 (7.61) | 13.05 (1.33) | 21.85 (30.90) |
| 4 × 4 | | | 13.12 (2.04) | 13.46 (13.00) | 30.87 (3.23) | 7.86 (4.36) | 20.95 (1.15) | 19.08 (4.01) | 13.11 (1.43) | 73.25 (12.25) |
| 8 × 8 | | | 14.10 (0.47) | 24.92 (34.63) | 33.05 (1.04) | 10.82 (5.21) | 23.07 (1.01) | 11.84 (4.84) | 15.27 (1.77) | 2.81 (0.66) |
| 16 × 16 | | | 14.56 (2.67) | 35.47 (36.63) | 32.77 (1.66) | 22.25 (4.21) | 23.35 (0.30) | 66.53 (10.35) | 15.54 (0.21) | 22.94 (32.37) |
| 32 × 32 | | | 16.25 (4.23) | 33.33 (47.14) | 33.22 (3.78) | 100.00 (0.00) | 26.03 (1.33) | 0.00 (0.00) | 18.15 (1.38) | 0.00 (0.00) |

Table 3: Defenses for simple-trigger on CIFAR-10 with distilled dataset size = 100.

For simple-trigger, we find that both CTA and ASR of None increase as we enlarge the trigger size. Moreover, both CTA and ASR of None increase as we enlarge the size of the distilled dataset. The above implies that simple-trigger is more suitable for large-size distilled datasets. Since the CTA and ASR increase as we enlarge the trigger, we focus on $32 \times 32$ trigger images in the following discussion. In the case of CIFAR-10, for size 100 (see Table 3), we can find that ASR of NAD is still 1. That is, NAD fails to remove the backdoor. For the other defenses, the CTA drops over $7\%$, though they can reduce the ASR. Hence, we conclude that these defenses are not effective. For size 500 (see Table 5 in Appendix A.8), the ASR of SCAn is still 1, implying that SCAn fails to remove the backdoor. The other defenses, SS, Strip, ABL, STRIP, and FP considerably compromise the CTA. Overall, the above results also suggest that the defenses may be more successful when we increase the size of the distilled dataset. On the other hand, for GTSRB (see Tabel 6 and Table 7 in Appendix A.8), we also reach a similar conclusion.

For relax-trigger (see Table 8 in Appendix A.8), all defenses considered in this paper cannot effectively remove the backdoor. In particular, in the case of CIFAR-10, for size 100, SCAn, AC, Strip, and ABL do not reduce the ASR. They even increase ASR to some degree. On the other hand, SS, STRIP, and FP also compromise the CTA too much. Lastly, though NAD reaches a better defense result; however, the corresponding ASR still remains about $50\%$ of None's ASR. Essentially, this suggests that NAD cannot completely defend against relax-trigger. For the other defenses, the ASR still remains over $30\%$ of None's ASR. These defenses are ineffective against relax-trigger.

In the case of GTSRB, for size 430, we can also find that SCAn, NAD, and STRIP cannot successfully remove the backdoor. The ASR still remains over $70\%$ of None's ASR. Besides, we can find that AC, SS, Stip, ABL, and FP still compromise the CTA too much. Finally, for size 2150, AC, Strip, NAD, and STRIP still remain ASR over $50\%$ of None's ASR. Furthermore, SCAn, ABL, and FP even increase the ASR. In addition, SS decreases the CTA by about $45\%$ of None's CTA. To sum up, relax-trigger shows strong backdoor resiliency against all the tested defenses.

## 5 CONCLUSION

In this paper, we present a novel theoretical framework based on the kernel inducing points (KIP) method to study the interplay between backdoor attacks and dataset distillation. The backdoor effect is characterized by three key components: conflict loss, projection loss, and generalization gap, along with two theory-induced attacks, simple-trigger and relax-trigger. Our simple-trigger proves that enlarged trigger size leads to improved ASR without sacrificing CTA. Our relax-trigger presents a new and resilient backdoor attack scheme that either completely breaks or significantly weakens eight existing backdoor defense methods. Our study provides novel theoretical insights, unveils new risks of dataset distillation-based backdoor attacks, and calls for better defenses.

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
