## A APPENDIX

### A.1 NOTATION TABLE

The notations used in this paper are presented in Table 4.

| Notations | Descriptions |
|---|---|
| $\mathcal{X}$ | feature space $\subset \mathbb{R}^d$ |
| $\mathcal{Y}$ | label space $\subset \mathbb{R}^c$ |
| $x$ | feature |
| $y$ | label |
| $\mathcal{D}$ | probability distribution which is distributed in $\mathcal{X} \times \mathcal{Y}$ |
| $\mathcal{D}_A$ | probability distribution which is distributed in $\mathcal{X} \times \mathcal{Y}$ for benign behaviors |
| $\mathcal{D}_B$ | probability distribution which is distributed in $\mathcal{X} \times \mathcal{Y}$ for malicious behavior (trigger) |
| $N$ | number of samples for some dataset |
| $N_A$ | number of samples for benign dataset |
| $N_B$ | number of samples for trigger dataset |
| $N_\mathcal{S}$ | number of samples for distilled dataset |
| $D$ | dataset picked from the distribution $\mathcal{D}$ with $N$ samples |
| $D_A$ | dataset picked from the distribution $\mathcal{D}_A$ with $N_A$ samples |
| $D_B$ | dataset picked from the distribution $\mathcal{D}_B$ with $N_B$ samples |
| $\mathcal{S}$ | any dataset with $N_\mathcal{S}$ samples |
| $\mathcal{S}^*$ | distilled dataset with $N_\mathcal{S}$ samples |
| $\mathcal{S}_A^*$ | distilled dataset from $D_A$ with $N_\mathcal{S}$ samples |
| $T$ | trigger pattern $\in \mathbb{R}^d$ |
| $y_T$ | trigger label $\in \mathcal{Y}$ |
| $\tilde{D}$ | poisoned dataset which is the union from $D_A$ and $D_B$ |
| $\boldsymbol{X}$ | the $N \times d$ matrix induced from the feature set in $D$. |
| $\boldsymbol{Y}$ | the $N \times c$ matrix induced from the label set in $D$. |
| $\boldsymbol{X}_A$ | the $N_A \times d$ matrix induced from the feature set in $D_A$. |
| $\boldsymbol{Y}_A$ | the $N_A \times c$ matrix induced from the label set in $D_A$. |
| $\boldsymbol{X}_B$ | the $N_B \times d$ matrix induced from the feature set in $D_B$. |
| $\boldsymbol{Y}_B$ | the $N_B \times c$ matrix induced from the label set in $D_B$. |
| $\boldsymbol{X}_\mathcal{S}$ | the $N_\mathcal{S} \times d$ matrix induced from the feature set in $\mathcal{S}$. |
| $\boldsymbol{Y}_\mathcal{S}$ | the $N_\mathcal{S} \times c$ matrix induced from the label set in $\mathcal{S}$. |
| $\boldsymbol{X}_{AB}$ | the $(N_A + N_B) \times d$ matrix induced from the feature set in $\tilde{D}$. |
| $\boldsymbol{Y}_{AB}$ | the $(N_A + N_B) \times c$ matrix induced from the label set in $\tilde{D}$. |
| $k(\cdot, \cdot)$ | the kernel |
| $\mathcal{H}_k$ | the reproducing kernel hilbert space induced by kernel $k$. |
| $\lambda$ | weight of regularization term. |
| $\lambda_\mathcal{S}$ | weight of regularization term for $\mathcal{S}$. |
| $\rho$ | penalty parameter. |
| $f_{\tilde{D}}$ | the model trained on $\tilde{D}$ with the weight of the regularization term $\lambda \geq 0$. |
| $f_\mathcal{S}$ | the model trained on $\mathcal{S}$ with the weight of the regularization term $\lambda_\mathcal{S} \geq 0$. |

Table 4: Notation Table

### A.2 LEMMA 1 AND ITS PROOF

**Lemma 1** (Projection lemma). *Given a synthetic dataset $\mathcal{S} = \{(x_s, y_s)\}_{s=1}^{N_\mathcal{S}}$, and a dataset $\tilde{D} = \{(x_i, y_i)\}_{i=1}^{N_A + N_B}$ where $(N_A + N_B)$ is the number of the samples of $\tilde{D}$. Suppose the kernel matrix*

$k(\boldsymbol{X}_{\mathcal{S}}, \boldsymbol{X}_{\mathcal{S}})$ is invertible, then we have

$$k(\cdot, x_i) = \underbrace{k(\cdot, \boldsymbol{X}_{\mathcal{S}})k(\boldsymbol{X}_{\mathcal{S}}, \boldsymbol{X}_{\mathcal{S}})^{-1}k(\boldsymbol{X}_{\mathcal{S}}, x_i)}_{\in \mathcal{H}_{\mathcal{S}}} \tag{22}$$

$$+ \underbrace{[k(\cdot, x_i) - k(\cdot, \boldsymbol{X}_{\mathcal{S}})k(\boldsymbol{X}_{\mathcal{S}}, \boldsymbol{X}_{\mathcal{S}})^{-1}k(\boldsymbol{X}_{\mathcal{S}}, x_i)]}_{\in \mathcal{H}_{\mathcal{S}}^{\perp}}, \quad \forall(x_i, y_i) \in \tilde{D} \tag{23}$$

where $\mathcal{H}_{\mathcal{S}} := span(\{k(\cdot, x_s) \in \mathcal{H}_k | (x_s, y_s) \in \mathcal{S}\})$ and $\mathcal{H}_{\mathcal{S}}^{\perp}$ is the collection of functions which is orthogonal to $\mathcal{H}_{\mathcal{S}}$ corresponding to the inner product $\langle \cdot, \cdot \rangle_{\mathcal{H}_k}$. The right hand side of (22) lies in $\mathcal{H}_{\mathcal{S}}$ while (23) lies in $\mathcal{H}_{\mathcal{S}}^{\perp}$. Thus, $k(\cdot, \boldsymbol{X}_{\mathcal{S}})k(\boldsymbol{X}_{\mathcal{S}}, \boldsymbol{X}_{\mathcal{S}})^{-1}k(\boldsymbol{X}_{\mathcal{S}}, x_i)$ is the solution of the optimization problem:

$$\arg\min_{f \in \mathcal{H}_S} \sum_{(x_s, y_s) \in \mathcal{S}} \|f(x_s) - k(x_s, x_i)\|_2^2. \tag{24}$$

*Proof.* $k(\cdot, \boldsymbol{X}_{\mathcal{S}})k(\boldsymbol{X}_{\mathcal{S}}, \boldsymbol{X}_{\mathcal{S}})^{-1}k(\boldsymbol{X}_{\mathcal{S}}, x_i)$ lies in $\mathcal{H}_{\mathcal{S}}$ is clearly. We just need to show that $k(\cdot, x_i) - k(\cdot, \boldsymbol{X}_{\mathcal{S}})k(\boldsymbol{X}_{\mathcal{S}}, \boldsymbol{X}_{\mathcal{S}})^{-1}k(\boldsymbol{X}_{\mathcal{S}}, x_i)$ lies in $\mathcal{H}_{\mathcal{S}}^{\perp}$. Notice that

$$\langle k(\cdot, x_s), k(\cdot, x_i) - k(\cdot, \boldsymbol{X}_{\mathcal{S}})k(\boldsymbol{X}_{\mathcal{S}}, \boldsymbol{X}_{\mathcal{S}})^{-1}k(\boldsymbol{X}_{\mathcal{S}}, x_i)) \rangle_{\mathcal{H}_k} \tag{25}$$

$$= k(x_s, x_i) - k(x_s, \boldsymbol{X}_{\mathcal{S}})k(\boldsymbol{X}_{\mathcal{S}}, \boldsymbol{X}_{\mathcal{S}})^{-1}k(\boldsymbol{X}_{\mathcal{S}}, x_i)), \quad \forall(x_s, y_s) \in \mathcal{S}. \tag{26}$$

If we collect all $\langle k(\cdot, x_s), k(\cdot, x_i) - k(\cdot, \boldsymbol{X}_{\mathcal{S}})k(\boldsymbol{X}_{\mathcal{S}}, \boldsymbol{X}_{\mathcal{S}})^{-1}k(\boldsymbol{X}_{\mathcal{S}}, x_i)) \rangle_{\mathcal{H}_k}$ for all $(x_s, y_s) \in \mathcal{S}$, we can obtain

$$k(\boldsymbol{X}_{\mathcal{S}}, x_i) - k(\boldsymbol{X}_{\mathcal{S}}, \boldsymbol{X}_{\mathcal{S}})k(\boldsymbol{X}_{\mathcal{S}}, \boldsymbol{X}_{\mathcal{S}})^{-1}k(\boldsymbol{X}_{\mathcal{S}}, x_i)) = k(\boldsymbol{X}_{\mathcal{S}}, x_i) - k(\boldsymbol{X}_{\mathcal{S}}, x_i) = 0. \tag{27}$$

This implies that $\langle k(\cdot, x_s), k(\cdot, x_i) - k(\cdot, \boldsymbol{X}_{\mathcal{S}})k(\boldsymbol{X}_{\mathcal{S}}, \boldsymbol{X}_{\mathcal{S}})^{-1}k(\boldsymbol{X}_{\mathcal{S}}, x_i)) \rangle_{\mathcal{H}_k} = 0$ for $x_s \in \mathcal{S}$. $k(\cdot, x_i) - k(\cdot, \boldsymbol{X}_{\mathcal{S}})k(\boldsymbol{X}_{\mathcal{S}}, \boldsymbol{X}_{\mathcal{S}})^{-1}k(\boldsymbol{X}_{\mathcal{S}}, x_i)$ lies in $\mathcal{H}_{\mathcal{S}}^{\perp}$. Eq. (27) also suggest that $k(x_s, \boldsymbol{X}_{\mathcal{S}})k(\boldsymbol{X}_{\mathcal{S}}, \boldsymbol{X}_{\mathcal{S}})^{-1}k(\boldsymbol{X}_{\mathcal{S}}, x_i)$ is equal to $k(x_s, x_i)$ for all $(x_s, y_s) \in \mathcal{S}$. So, $k(\cdot, \boldsymbol{X}_{\mathcal{S}})k(\boldsymbol{X}_{\mathcal{S}}, \boldsymbol{X}_{\mathcal{S}})^{-1}k(\boldsymbol{X}_{\mathcal{S}}, x_i)$ is the solution of Eq. (24). $\blacksquare$

### A.3 THEOREM 1 AND ITS PROOF

**Theorem 1** (Upper bound of conflict loss). *The conflict loss $\mathcal{L}_{conflict}$ can be bounded as*

$$\mathcal{L}_{conflict} \le \frac{1}{N_A + N_B} Tr(\boldsymbol{I} - k(\boldsymbol{X}_{AB}, \boldsymbol{X}_{AB})[k(\boldsymbol{X}_{AB}, \boldsymbol{X}_{AB}) + (N_A + N_B)\lambda\boldsymbol{I}]^{-1})^2)\|\boldsymbol{Y}_{AB}\|_2^2 \tag{28}$$

*where Tr is the trace operator, $k(\boldsymbol{X}_{AB}, \boldsymbol{X}_{AB})$ is a $(N_A + N_B) \times (N_A + N_B)$ matrix, and $\boldsymbol{Y}_{AB}$ is a $(N_A + N_B) \times c$ matrix.*

*Proof.* From Definition 1, we know that the kernel matrix $k(\boldsymbol{X}_{AB}, \boldsymbol{X}_{AB})$ is positive semidefinite. Hence, there exist some unitary matrix $\boldsymbol{U}$ such that $k(\boldsymbol{X}_{AB}, \boldsymbol{X}_{AB}) = \boldsymbol{U}\Sigma\boldsymbol{U}^T$ where $\Sigma$ is some diagonal matrix with non-negative components. Then, from Eq. (12), we can express the upper bound of the conflict loss $\mathcal{L}_{conflict}$ as

$$\mathcal{L}_{conflict} = \frac{1}{N_A + N_B}\|\boldsymbol{I}\boldsymbol{Y}_{AB} - \boldsymbol{U}\Sigma\boldsymbol{U}^T[\boldsymbol{U}\Sigma\boldsymbol{U}^T + (N_A + N_B)\lambda\boldsymbol{I}]^{-1}\boldsymbol{Y}_{AB}\|_2^2 \tag{29}$$

$$= \frac{1}{N_A + N_B}\|\boldsymbol{U}\boldsymbol{I}\boldsymbol{U}^T\boldsymbol{Y}_{AB} - \boldsymbol{U}\Sigma\boldsymbol{U}^T[\boldsymbol{U}(\Sigma + (N_A + N_B)\lambda\boldsymbol{I})\boldsymbol{U}^T]^{-1}\boldsymbol{Y}_{AB}\|_2^2 \tag{30}$$

$$= \frac{1}{N_A + N_B}\|\boldsymbol{U}(\boldsymbol{I} - \Sigma[\Sigma + (N_A + N_B)\lambda\boldsymbol{I})]^{-1})\boldsymbol{U}^T\boldsymbol{Y}_{AB}|_2^2 \tag{31}$$

$$= \frac{1}{N_A + N_B}\|(\boldsymbol{I} - \Sigma[\Sigma + (N_A + N_B)\lambda\boldsymbol{I})]^{-1})\boldsymbol{U}^T\boldsymbol{Y}_{AB}|_2^2 \tag{32}$$

$$\le \frac{1}{N_A + N_B}\|Tr(\boldsymbol{I} - \Sigma[\Sigma + (N_A + N_B)\lambda\boldsymbol{I})]^{-1})\boldsymbol{U}^T\boldsymbol{Y}_{AB}|_2^2 \tag{33}$$

$$= \frac{1}{N_A + N_B}Tr(\boldsymbol{I} - \Sigma[\Sigma + (N_A + N_B)\lambda\boldsymbol{I})]^{-1})^2\|\boldsymbol{Y}_{AB}\|_2^2. \tag{34}$$

Moreover, we have

$$\text{Tr}(\boldsymbol{I} - k(\boldsymbol{X}_{AB}, \boldsymbol{X}_{AB})[k(\boldsymbol{X}_{AB}, \boldsymbol{X}_{AB}) + (N_A + N_B)\lambda \boldsymbol{I}]^{-1})$$

$$= \text{Tr}(\boldsymbol{U}(\boldsymbol{I} - \Sigma[\Sigma + (N_A + N_B)\lambda \boldsymbol{I}]^{-1})\boldsymbol{U}^T) \tag{35}$$

$$= \text{Tr}((\boldsymbol{I} - \Sigma[\Sigma + (N_A + N_B)\lambda \boldsymbol{I}]^{-1})\boldsymbol{U}^T\boldsymbol{U}) \tag{36}$$

$$= \text{Tr}((\boldsymbol{I} - \Sigma[\Sigma + (N_A + N_B)\lambda \boldsymbol{I}]^{-1})). \tag{37}$$

Combining Eq. (34) and Eq. (37) completes the proof. ∎

### A.4 THEOREM 2 AND ITS PROOF

**Theorem 2** (Upper bound of projection loss). *Suppose the kernel matrix of the synthetic dataset $k(\boldsymbol{X}_\mathcal{S}, \boldsymbol{X}_\mathcal{S})$ is invertible, $f_\mathcal{S}$ is the model trained on the synthetic dataset $\mathcal{S}$ with the regularization term $\lambda_\mathcal{S}$, where the projection loss $\mathcal{L}_{project} = \min_S \mathbb{E}_{(x,y)\sim\tilde{D}}\ell(f_S, (x, f_{\tilde{D}}(x)))$ can be bounded as*

$$\mathcal{L}_{project} \leq \sum_{(x_i, y_i)\in\tilde{D}} \min_{\boldsymbol{X}_\mathcal{S}} \sum_{j=1}^{c} \frac{|\alpha_{i,j}|^2}{N_A + N_B} \|k(\boldsymbol{X}_{AB}, x_i) - k(\boldsymbol{X}_{AB}, \boldsymbol{X}_\mathcal{S})k(\boldsymbol{X}_\mathcal{S}, \boldsymbol{X}_\mathcal{S})^{-1}k(\boldsymbol{X}_\mathcal{S}, x_i)\|_2^2.$$
$$\tag{38}$$

*where $\alpha_{i,j} := [[k(\boldsymbol{X}_{AB}, \boldsymbol{X}_{AB}) + (N_A + N_B)\lambda \boldsymbol{I}]^{-1}\boldsymbol{Y}_{AB}]_{i,j}$, which is the weight of $k(\cdot, x_i)$ corresponding to $f_{\tilde{D}}^j$, $\boldsymbol{X}_{AB}$ is the $(N_A + N_B) \times d$ matrix corresponding to the features of $\tilde{D}$, $\boldsymbol{X}_\mathcal{S}$ is the $N_\mathcal{S} \times d$ matrix corresponding to the features of $\mathcal{S}$, $\boldsymbol{Y}_{AB}$ is the $(N_A + N_B) \times c$ matrix corresponding to the labels of $\tilde{D}$, $\boldsymbol{Y}_\mathcal{S}$ is the $N_\mathcal{S} \times c$ matrix corresponding to the labels of $\mathcal{S}$.*

*Proof.* From (11), we know that

$$f_{\tilde{D}}^j(x) = [k(x, \boldsymbol{X}_{AB})[k(\boldsymbol{X}_{AB}, \boldsymbol{X}_{AB}) + (N_A + N_B)\lambda \boldsymbol{I}]^{-1}\boldsymbol{Y}_{AB}]_j$$

$$= \sum_{(x_i, y_i)\in\tilde{D}} \alpha_{i,j}k(x, x_i). \tag{39}$$

Then, we can bound the projection loss as

$$\mathcal{L}_{\text{project}} = \min_\mathcal{S} \mathbb{E}_{(x,y)\sim\tilde{D}}\ell(f_\mathcal{S}, (x, f_{\tilde{D}}(x)))$$

$$= \min_\mathcal{S} \frac{1}{N_A + N_B} \sum_{(x,y)\in\tilde{D}} \ell(f_\mathcal{S}, (x, f_{\tilde{D}}(x))) \tag{40}$$

$$\leq \sum_{(x_i, y_i)\in\tilde{D}} \min_\mathcal{S} \left\{ \frac{1}{N_A + N_B} \sum_{(x,y)\in\tilde{D}} \sum_{j=1}^{c} \ell(f_\mathcal{S}^j, (x, \alpha_{i,j}k(x, x_i))) \right\} \tag{41}$$

$$= \sum_{(x_i, y_i)\in\tilde{D}} \min_\mathcal{S} \left\{ \frac{1}{N_A + N_B} \sum_{(x,y)\in\tilde{D}} \sum_{j=1}^{c} \|[k(x, \boldsymbol{X}_\mathcal{S})[k(\boldsymbol{X}_\mathcal{S}, \boldsymbol{X}_\mathcal{S}) + N_\mathcal{S}\lambda_\mathcal{S}\boldsymbol{I}]^{-1}\boldsymbol{Y}_\mathcal{S}]_j - \alpha_{i,j}k(x, x_i)\|_2^2 \right\}. \tag{42}$$

For each $(x_i, y_i) \in \tilde{D}$, we have

$$\min_\mathcal{S} \left\{ \frac{1}{N_A + N_B} \sum_{(x,y)\in\tilde{D}} \sum_{j=1}^{c} \|[k(x, \boldsymbol{X}_\mathcal{S})[k(\boldsymbol{X}_\mathcal{S}, \boldsymbol{X}_\mathcal{S}) + N_\mathcal{S}\lambda_\mathcal{S}\boldsymbol{I}]^{-1}\boldsymbol{Y}_\mathcal{S}]_j - \alpha_{i,j}k(x, x_i)\|_2^2 \right\}$$

$$\leq \min_{\boldsymbol{X}_\mathcal{S}} \left\{ \frac{1}{N_A + N_B} \sum_{(x,y)\in\tilde{D}} \sum_{j=1}^{c} \min_{\boldsymbol{Y}_\mathcal{S}} \|[k(x, \boldsymbol{X}_\mathcal{S})[k(\boldsymbol{X}_\mathcal{S}, \boldsymbol{X}_\mathcal{S}) + N_\mathcal{S}\lambda_\mathcal{S}\boldsymbol{I}]^{-1}\boldsymbol{Y}_\mathcal{S}]_j - \alpha_{i,j}k(x, x_i)\|_2^2 \right\}$$
$$\tag{43}$$

$$= \min_{\boldsymbol{X}_\mathcal{S}} \left\{ \frac{1}{N_A + N_B} \sum_{(x,y)\in\tilde{D}} \sum_{j=1}^{c} \min_{f_{i,j}\in\mathcal{H}_\mathcal{S}} \|f_{i,j}(x) - \alpha_{i,j}k(x, x_i)\|_2^2 \right\}. \tag{44}$$

Then, with the help of Lemma 1, we bound Eq. (44) as follows

$$
\min_{\boldsymbol{X}_S} \left\{ \frac{1}{N_A + N_B} \sum_{(x,y) \in \tilde{D}} \sum_{j=1}^{c} \min_{f_{i,j} \in \mathcal{H}_S} \|f_{i,j}(x) - \alpha_{i,j} k(x, x_i)\|_2^2 \right\}
$$

$$
\leq \min_{\boldsymbol{X}_S} \left\{ \frac{1}{N_A + N_B} \sum_{(x,y) \in \tilde{D}} \sum_{j=1}^{c} \|\alpha_{i,j}[k(x, x_i) - k(x, \boldsymbol{X}_S)k(\boldsymbol{X}_S, \boldsymbol{X}_S)^{-1} k(\boldsymbol{X}_S, x_i)]\|_2^2 \right\} \quad (45)
$$

$$
\leq \min_{\boldsymbol{X}_S} \left\{ \sum_{j=1}^{c} \frac{|\alpha_{i,j}|^2}{N_A + N_B} \|k(\boldsymbol{X}_{AB}, x_i) - k(\boldsymbol{X}_{AB}, \boldsymbol{X}_S)k(\boldsymbol{X}_S, \boldsymbol{X}_S)^{-1} k(\boldsymbol{X}_S, x_i)\|_2^2 \right\} \quad (46)
$$

We take the summation over $(x_i, y_i) \in \tilde{D}$ for Eq. (46) and then derive the upper bound. ∎

## A.5 THEOREM 3 AND ITS PROOF

**Theorem 3** (Upper bound of generalization gap). *Given a $N$-sample dataset $D$, sampled from the distribution $\mathcal{D}$, then the following generalization gap holds for all $g \in \mathcal{G}$ with probability at least $1 - \delta$:*

$$
\mathbb{E}_{(x,y) \sim \mathcal{D}} g((x,y)) - \sum_{(x_i, y_i) \in D} \frac{g((x_i, y_i))}{N} \leq 2\hat{\mathfrak{R}}_D(\mathcal{G}) + 3L_{\mathcal{D}}\Gamma_{\mathcal{D}} \sqrt{\frac{\log \frac{2}{\delta}}{2N}}, \quad (47)
$$

*where $\boldsymbol{X}$ is the matrix corresponding to the features of $D$ and $\hat{\mathfrak{R}}_D(\mathcal{G})$ is the empirical Rademacher's complexity.*

*Proof.* Here we only sketch the proof, which mainly follows the proof of Theorem 3.3 in (Mohri et al., 2012), but is slightly modified under our assumption. First, we denote the maximum of the generalization gap for the dataset $D$ as

$$
\Phi(D) = \sup_{g \in \mathcal{G}} (\mathbb{E}_{(x,y) \in \mathcal{D}} g((x,y)) - \frac{1}{N} \sum_{(x_i, y_i) \in D} g((x_i, y_i))). \quad (48)
$$

Consider another dataset $D'$ sampled from the distribution $\mathcal{D}$. $D$ and $D'$ differ by only one sample, which is denoted as $(x_N, y_N)$ and $(x'_N, y'_N)$. Then, according to our assumption, we have

$$
\Phi(D) - \Phi(D') \leq \sup_{g \in \mathcal{G}} (\frac{1}{N} g((x_N, y_N)) - \frac{1}{N} g((x'_N, y'_N))) \quad (49)
$$

$$
\leq \frac{L_D \|(x_N, y_N) - x'_N, y'_N)\|_2}{N} \quad (50)
$$

$$
\leq \frac{L_{\mathcal{D}}\Gamma_{\mathcal{D}}}{N}. \quad (51)
$$

Then, we can apply McDiarmid's inequality on $\Phi(D)$. We can derive

$$
\Phi(D) \leq \mathbb{E}_D \Phi(D) + L_{\mathcal{D}}\Gamma_{\mathcal{D}} \sqrt{\frac{\log \frac{2}{\delta}}{2N}}, \quad (52)
$$

which holds with probability at least $1 - \frac{\delta}{2}$. In the proof of Theorem 3.3 in (Mohri et al., 2012), we can also prove that $\mathbb{E}_D \Phi(D) \leq 2\mathfrak{R}(\mathcal{G})$, where $\mathfrak{R}(\mathcal{G})$ is Rademacher's complexity. Under our assumption, we notice that the empirical Rademacher complexity $\hat{\mathfrak{R}}_D(\mathcal{G})$ also satisfies

$$
\hat{\mathfrak{R}}_D(\mathcal{G}) - \hat{\mathfrak{R}}_{D'}(\mathcal{G}) \leq \frac{L_{\mathcal{D}}\Gamma_{\mathcal{D}}}{N}. \quad (53)
$$

So, we can apply McDiarmid's inequality again and obtain

$$
\mathfrak{R}(\mathcal{G}) \leq \hat{\mathfrak{R}}_D(\mathcal{G}) + L_{\mathcal{D}}\Gamma_{\mathcal{D}} \sqrt{\frac{\log \frac{2}{\delta}}{2N}}, \quad (54)
$$

which holds with probability at least $1 - \frac{\delta}{2}$. Combine (52), (54) and the fact that $\mathbb{E}_D \Phi(D) \leq 2\Re(\mathcal{G})$, we have

$$\mathbb{E}_{(x,y) \sim \mathcal{D}} g((x,y)) - \sum_{(x_i, y_i) \in D} \frac{g((x_i, y_i))}{N} \leq 2\hat{\Re}_D(\mathcal{G}) + 3L_{\mathcal{D}}\Gamma_{\mathcal{D}} \sqrt{\frac{\log \frac{2}{\delta}}{2N}}. \tag{55}$$

which holds with probability at least $1 - \delta$. ∎

## A.6 Pseudocode for The Simplest Form of KIP-based Backdoor Attack

---
**Algorithm 1** The Simplest Form of KIP-based Backdoor Attack
---
**Require:** benign dataset $D_A$, initial trigger $T_0$, trigger label $y_T$, mask $m$, size of distilled dataset $N_{\mathcal{S}}$, training step STEP $> 0$, batch size BATCH $> 0$, mix ratio $\rho_m > 0$, learning rate $\eta > 0$.
**Ensure:** synthetic dataset $\mathcal{S}^*$
    $N \leftarrow 1$
    $\mathcal{S} \leftarrow$ Randomly sample $N_{\mathcal{S}}$ data from $D_A$ as initial distilled dataset.
    $D_B \leftarrow \{(x_b, y_b) := ((1-m) \odot x + m \odot T, y_T) | (x_a, y_a) \in D_A\}$
    **while** $N \leq$ STEP **do**
        $(\boldsymbol{X}_A^{\text{batch}}, \boldsymbol{Y}_A^{\text{batch}}) \leftarrow$ Randomly sample BATCH data from $D_A$.
        $(\boldsymbol{X}_B^{\text{batch}}, \boldsymbol{Y}_B^{\text{batch}}) \leftarrow$ Randomly sample BATCH data from $D_B$.
        $\tilde{D}^{\text{batch}} \leftarrow (\boldsymbol{X}_A^{\text{batch}}, \boldsymbol{Y}_A^{\text{batch}}) \cup (\boldsymbol{X}_B^{\text{batch}}, \boldsymbol{Y}_B^{\text{batch}})$
        $\mathcal{S} \leftarrow \mathcal{S} - \eta \nabla_{\mathcal{S}} \mathcal{L}(\mathcal{S},)$            $\triangleright$ $\mathcal{L}$ is defined in Eq. (8).
        $N \leftarrow N + 1$
    **end while**
    $\mathcal{S}^* \leftarrow \mathcal{S}$
---

## A.7 Pseudocode for relax-trigger

---
**Algorithm 2** relax-trigger
---
**Require:** benign dataset $D_A$, initial trigger $T_0$, trigger label $y_T$, mask $m$, training step STEP $> 0$, batch size BATCH $> 0$, mix ratio $\rho_m > 0$, penalty parameter $\rho > 0$, learning rate $\eta > 0$.
**Ensure:** optimized $T^*$
    $T \leftarrow T_0$
    $N \leftarrow 1$
    $\mathcal{S}_A^* \leftarrow$ Apply KIP to $D_A$            $\triangleright$ We use $\mathcal{S}_A^*$ to denote $\mathcal{S}^*$ from $D_A$
    **while** $N \leq$ STEP **do**
        $(\boldsymbol{X}_A^{\text{batch}}, \boldsymbol{Y}_A^{\text{batch}}) \leftarrow$ Randomly pick BATCH samples from $D_A$
        $(\boldsymbol{X}^{\text{batch}}, \boldsymbol{Y}^{\text{batch}}) \leftarrow$ Randomly pick BATCH $\times \rho_m$ samples from $D_A$
        $(\boldsymbol{X}_B^{\text{batch}}, \boldsymbol{Y}_B^{\text{batch}}) \leftarrow \{(x_b, y_b) := ((1-m) \odot x + m \odot T, y_T) | (x, y) \in (\boldsymbol{X}^{\text{batch}}, \boldsymbol{Y}^{\text{batch}})\}$
        $T \leftarrow T - \eta \nabla_T \mathcal{L}(\mathcal{S}_A^*, (\boldsymbol{X}_A^{\text{batch}}, \boldsymbol{Y}_A^{\text{batch}}), (\boldsymbol{X}_B^{\text{batch}}, \boldsymbol{Y}_B^{\text{batch}}), \rho)$     $\triangleright$ $\mathcal{L}$ is defined in Eq. (21).
        $N \leftarrow N + 1$
    **end while**
    $T^* \leftarrow T$
---

## A.8 Extra Experiments.

In Tables 5 ∼ 8, we provide extra experimental results.

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

Table 8: Defenses for relax-trigger on CIFAR-10 and GTSRB.

# B  ABLATION STUDIES

## B.1  KIP-BASED BACKDOOR ATTACK ON IMAGENET

We perform our KIP-based backdoor attack on ImageNet. In our experiment, we randomly choose ten sub-classes to perform our experiment. We also resize each image in the ImageNet into 128x128. The experimental results show that our KIP-based backdoor attack is effective (see Table 9).

## B.2  IMPACT OF IPC ON KIP-BASED BACKDOOR ATTACK

We examine the efficacy of KIP-based backdoor attack influenced by IPC (Image Per Class). We examine the efficacy of simple-trigger and relax-trigger under different sizes of synthetic dataset (IPC 10 ∼ IPC 50). The experimental results show that both CTA and ASR are gradually rising as

| Trigger-type | Dataset | Model | IPC (Image Per Class) | CTA (%) | ASR (%) |
|---|---|---|---|---|---|
| simple-trigger | ImageNet | NTK | 10 | 15.00 | 100.00 |
| simple-trigger | ImageNet | NTK | 50 | 16.60 | 100.00 |
| relax-trigger | ImageNet | NTK | 10 | 16.40 | 100.00 |
| relax-trigger | ImageNet | NTK | 50 | 17.00 | 100.00 |

Table 9: Efficacy of KIP-based backdoor attack on ImageNet.

the IPC increases (see Table 10). The corresponding experiments for DOORPING is presented in Table 11.

| Dataset | Trigger-type | IPC (Image Per Class) | CTA (%) | ASR (%) |
|---|---|---|---|---|
| CIFAR-10 | simple-trigger | 10 | 41.70 (0.25) | 100 (0.00) |
| CIFAR-10 | simple-trigger | 20 | 42.58 (0.23) | 100 (0.00) |
| CIFAR-10 | simple-trigger | 30 | 43.29 (0.35) | 100 (0.00) |
| CIFAR-10 | simple-trigger | 40 | 43.55 (0.42) | 100 (0.00) |
| CIFAR-10 | simple-trigger | 50 | 43.66 (0.40) | 100 (0.00) |
| | | | | |
| CIFAR-10 | relax-trigger | 10 | 41.66 (0.01) | 100 (0.00) |
| CIFAR-10 | relax-trigger | 20 | 42.46 (0.01) | 100 (0.00) |
| CIFAR-10 | relax-trigger | 30 | 42.99 (0.08) | 100 (0.00) |
| CIFAR-10 | relax-trigger | 40 | 43.10 (0.09) | 100 (0.00) |
| CIFAR-10 | relax-trigger | 50 | 43.64 (0.40) | 100 (0.00) |
| | | | | |
| GTSRB | simple-trigger | 10 | 67.56 (0.60) | 100 (0.00) |
| GTSRB | simple-trigger | 20 | 69.44 (0.35) | 100 (0.00) |
| GTSRB | simple-trigger | 30 | 70.24 (0.38) | 100 (0.00) |
| GTSRB | simple-trigger | 40 | 70.84 (0.32) | 100 (0.00) |
| GTSRB | simple-trigger | 50 | 71.27 (0.24) | 100 (0.00) |
| | | | | |
| GTSRB | relax-trigger | 10 | 68.73 (0.67) | 95.26 (0.54) |
| GTSRB | relax-trigger | 20 | 70.38 (0.03) | 94.85 (0.13) |
| GTSRB | relax-trigger | 30 | 71.26 (0.02) | 95.73 (0.32) |
| GTSRB | relax-trigger | 40 | 71.81 (0.01) | 95.84 (0.18) |
| GTSRB | relax-trigger | 50 | 71.54 (0.33) | 95.08 (0.33) |

Table 10: Efficacy of KIP-based backdoor attack influenced by the size of the synthetic dataset.

| Dataset | Trigger-type | IPC (Image Per Class) | CTA (%) | ASR (%) |
|---|---|---|---|---|
| CIFAR-10 | DOORPING | 10 | 36.35 (0.42) | 80.00 (40.00) |
| CIFAR-10 | DOORPING | 20 | 37.65 (0.42) | 70.00 (45.83) |
| CIFAR-10 | DOORPING | 30 | 38.48 (0.36) | 90.00 (30.00) |
| CIFAR-10 | DOORPING | 40 | 37.78 (0.61) | 70.00 (45.83) |
| GTSRB | DOORPING | 10 | 68.03 (0.92) | 90.00 (30.00) |
| GTSRB | DOORPING | 20 | 81.45 (0.46) | 80.00 (40.00) |
| GTSRB | DOORPING | 30 | 81.62 (0.71) | 100.00 (0.00) |

Table 11: Efficacy of DOORPING influenced by the size of the synthetic dataset.

## B.3 CROSS MODEL ABILITY OF KIP-BASED BACKDOOR ATTACK

The experiment for cross model ability is presented in Table 12. We train the distilled dataset poinsoned by simple-trigger and relax-trigger on 3-layers MLP and 3-layers ConvNet. The experimental results show that both CTA and ASR go up as we increase the IPC (Image Per Class), which suggests that the cross model issue may be relieved as the IPC is large enough.

| Dataset | Trigger-type | IPC (Image Per Class) | Cross_model | CTA (%) | ASR(%) |
|---------|--------------|------------------------|-------------|---------|--------|
| CIFAR-10 | simple-trigger | 10 | MLP | 11.58 (2.10) | 40.00 (48.98) |
| CIFAR-10 | simple-trigger | 10 | CNN | 47.37 (7.44) | 40.00 (48.98) |
| CIFAR-10 | simple-trigger | 10 | NTK (baseline) | 41.70 (0.25) | 100.00 (0.00) |
| | | | | | |
| CIFAR-10 | simple-trigger | 50 | MLP | 48.08 (4.72) | 40.00 (48.98) |
| CIFAR-10 | simple-trigger | 50 | CNN | 95.96 (1.10) | 100.00 (0.00) |
| CIFAR-10 | simple-trigger | 50 | NTK (baseline) | 43.66 (0.40) | 100.00 (0.00) |
| | | | | | |
| CIFAR-10 | relax-trigger | 10 | MLP | 10.52 (7.44) | 19.40 (38.80) |
| CIFAR-10 | relax-trigger | 10 | CNN | 64.21 (6.98) | 81.80 (11.44) |
| CIFAR-10 | relax-trigger | 10 | NTK (baseline) | 41.66 (0.74) | 100.00 (0.00) |
| | | | | | |
| CIFAR-10 | relax-trigger | 50 | MLP | 44.24 (4.49) | 78.28 (24.57) |
| CIFAR-10 | relax-trigger | 50 | CNN | 93.13 (2.24) | 82.80 (6.53) |
| CIFAR-10 | relax-trigger | 50 | NTK (baseline) | 43.64 (0.40) | 100.00 (0.00) |

Table 12: Experiment of cross model ability of KIP-based backdoor attack.

### B.4 TRANSFERABILITY OF KIP-BASED BACKDOOR ATTACK

Our KIP-based backdoor attack can evade other data distillation techniques. In particular, we perform experiments to examine the transferability of our theory-induced triggers. We first use our simple-trigger and relax-trigger to poison the dataset. Then, we distill dataest with a different distillation method, FRePo (Zhou et al., 2022) and DM (Zhao & Bilen, 2023). The experimental results shows that our triggers can successfully transfer to the FrePo and DM (see Table 13 and Table 14).

| Trigger-type | Dataset | IPC (Image Per Class) | Distillation | Model | CTA (%) | ASR (%) |
|--------------|---------|------------------------|--------------|-------|---------|---------|
| CIFAR-10 | simple-trigger | 10 | FRePO | ConvNet | 60.32 | 83.10 |
| CIFAR-10 | relax-trigger | 50 | FRePO | ConvNet | 68.34 | 81.61 |

Table 13: Experiment of transferability for FRePO.

| Trigger-type | Dataset | IPC (Image Per Class) | Distillation | Model | CTA (%) | ASR (%) |
|--------------|---------|------------------------|--------------|-------|---------|---------|
| Cifar10 | simple-trigger | 10 | DM | MLP | 36.41 | 77.03 |
| Cifar10 | simple-trigger | 50 | DM | MLP | 36.88 | 76.79 |
| Cifar10 | relax-trigger | 10 | DM | MLP | 36.31 | 76.04 |
| Cifar10 | relax-trigger | 50 | DM | MLP | 36.81 | 76.21 |

Table 14: Experiment of transferability for DM.

### B.5 KIP-BASED BACKDOOR ATTACK ON NAS AND CL

We train our distilled dataset poinsoned by simple-trigger and relax-trigger in different scenarios, neural architecture search (NAS) and continual learning (CL). The experimental results are shown in Table 15 and Table 16.

| Trigger-type | Dataset | IPC (Image Per Class) | Scenario | CTA (%) | ASR (%) |
|--------------|---------|------------------------|----------|---------|---------|
| simple-trigger | CIFAR-10 | 50 | NAS | 37.49(3.44) | 100.00(0.00) |
| relax-trigger | CIFAR-10 | 50 | NAS | 36.43(3.62) | 86.23(3.22) |

Table 15: Experiment for NAS. The experiment result shows that our triggers remain effective for NAS.

Note that the details about our implementation of NAS and CL are described below.

| Trigger-type | Dataset | IPC (Image Per Class) | Scenario | CTA (%) | ASR (%) |
|---|---|---|---|---|---|
| simple-trigger | CIFAR-10 | 50 | CL | 13.93(1.93) | 100.00(0.00) |
| simple-trigger | CIFAR-10 | 50 | baseline | 13.60(1.66) | 100.00(0.00) |
| relax-trigger | CIFAR-10 | 50 | CL | 20.13(2.94) | 60.94(21.68) |
| relax-trigger | CIFAR-10 | 50 | baseline | 14.00(3.54) | 43.11(7.83) |

Table 16: Experiment for CL. The experiment result shows that both CTA and ASR are slightly higher than baseline.

**NAS** : The process defines a search space (random search) that includes a range of possible model parameters such as the number of convolutional layers, the number of dense layers, and the size of the convolutional layers. The program randomly selects parameters from this space to generate multiple candidate model architectures. A CNN model is then built, comprising convolutional layers (Conv2D), batch normalization (BatchNormalization), activation functions (such as ReLU), pooling layers (MaxPooling2D), flattening layers (Flatten), fully connected layers (Dense), and optionally Dropout layers. Each model is compiled and trained using the Adam optimizer and categorical cross-entropy loss function, but in this case, the same dataset is used for evaluation (although typically, an independent test set should be used). The accuracy and loss functions of different models are compared, and ultimately the best model is selected and saved

**CL** : The dataset is divided into different category-specific subsets (as in CIFAR-10, which is divided into 10 categories), each containing images and their corresponding labels. This allows the model to gradually train on each subset. A CNN model is built, including multiple convolutional layers (Conv2D), batch normalization layers (BatchNormalization), ReLU activation functions, max pooling layers (MaxPooling2D), and fully connected layers (Dense). The final layer uses a softmax activation function, a typical configuration for label classification tasks. The model is compiled using an RMSprop optimizer and categorical cross-entropy loss function. Further training optimization can be applied, such as using Elastic Weight Consolidation (EWC) to minimize the impact on the originally trained model when learning new subsets.

### B.6    PERFORMANCE OF THE TRIGGERS WITHOUT DISTILLATION

We perform the experiments on CIFAR-10 and GTSRB. We first utilize the simple-trigger and relax-trigger to poison the dataset. Then, we use 3-layers ConvNet to train a model and evaluate corresponding CTA and ASR. The experimental results demonstrate that our triggers simple-trigger and relax-trigger both remain effective (see Table 17).

| Dataset | Trigger-type | Transparency (m) | CTA (%) | ASR (%) |
|---|---|---|---|---|
| CIFAR-10 | simple-trigger | 1 | 70.02 (0.40) | 100.00 (0.00) |
| CIFAR-10 | relax-trigger | 0.3 | 70.02 (0.65) | 99.80 (0.04) |
| CIFAR-10 | simple-trigger | 0.3 | 67.84 (0.36) | 95.50 (1.23) |
| | | | | |
| GTSRB | simple-trigger | 1 | 72.47 (3.36) | 100.00 (0.00) |
| GTSRB | relax-trigger | 0.3 | 75.50 (2.09) | 99.82 (0.09) |
| GTSRB | simple-trigger | 0.3 | 70.21 (3.03) | 99.36 (0.20) |

Table 17: Experiment of the performance of the triggers without distillation.