# OpenReview forum: "Rethinking Backdoor Attacks on Dataset Distillation: A Kernel Method Perspective"
_ICLR.cc/2024/Conference — ICLR 2024 poster_

### Official Review · Reviewer_bXcG · 2023-10-26

**Soundness:** 2 fair
**Presentation:** 2 fair
**Contribution:** 2 fair
**Rating:** 5
**Confidence:** 4

**Summary:**

This paper study the backdoor attack for the Kernel Inducing Points (KIP) based dataset distillation. Specifically, the paper introduces two theory-driven trigger patterns and provides empirical evidence that they can increase ASR of models (with the same architecture as the proxies for dataset distillation) trained on the distilled datasets without sacrificing CTA remarkably. Additionally, experimental results also indicate that the evaluated backdoor defense methods may not be fully effective against the proposed relax-trigger.

**Strengths:**

S1: This paper proposes to investigate a theoretical framework for the KIP-based dataset distillation method about why certain backdoors can survive in distilled datasets. Then, two theory-driven backdoor trigger patterns are consequently introduced.

S2: In the evaluated scenarios, the proposed triggers present adequate ability to raise ASR without sacrificing CTA remarkably. Moreover, experimental results also indicate that the 8 evaluated backdoor defence methods may not be fully effective against the proposed relax-trigger.

**Weaknesses:**

W1: Certain key aspects of the presented theory appear ambiguous from my vantage point. I will delve deeper into these ambiguities in the Questions section.

W2: The established theoretical framework predominantly caters to kernel-based dataset distillation methods, and its application seems restricted primarily to the initially proposed KIP method. Given the complexities associated with computing the NTK, KIP's practicality has been empirically questioned. While subsequent kernel-based dataset distillation methods capable of distilling comprehensive datasets have emerged (e.g., [1]), this paper falls short of validating their compatibility with the introduced framework. This oversight not only raises concerns about the paper's soundness but also limits the practicability of the introduced attack method.

W3: The experimental design in the paper appears insufficient, raising questions about the broader applicability of the proposed attack. First, it relies on a mere two benchmark datasets. Second, the distilled dataset's size variation is limited to IPC (abbreviation of Image Per Class) scenarios of 10 and 50. Third, while cross-architecture generalization is pivotal in dataset distillation, the paper's evaluations seem to be confined to a 3-layer ConvNet, which is consistent with the architecture of the proxy model designated for distillation.

[1] Yongchao Zhou, Ehsan Nezhadarya, Jimmy Ba: Dataset Distillation using Neural Feature Regression. NeurIPS 2022

**Questions:**

Q1: Regarding Equation 9, why is the objective of the KIP-based backdoor attack to minimize the empirical loss of $f_{\mathcal{S}^*}$ on either $\mathcal{D}_A$ or $\mathcal{D}_B$, rather than simultaneously reducing the empirical loss on both $\mathcal{D}_A$ and $\mathcal{D}_B\$ as your statement about "Backdoor Attack" (The next to the last paragraph above Equation 7)?

While you attempt to address this in Equation 10 by introducing $\tilde{D}=D_A \cup D_B$ to establish an upper bound on the loss of $f_{\mathcal{S}^*}$ with respect to $D$, this formulation seems somewhat unreasonable to me.

Q2: It appears that the introduced projection loss can be directly optimized with respect to the trigger $T$. What's the rationale behind setting an upper bound and optimizing the projection loss through this bound? Does this approach offer computational benefits?

Q3: Based on my W3, could you share additional experimental evidence to validate the efficacy of your proposed triggers when applied to models with alternative architectures trained on the synthesized datasets?

---

> ### Author Response · Authors · 2023-11-21
>
> # Reviewer bXcG
> ## Weakness
> ### W1. Certain key aspects of the presented theory appear ambiguous from my vantage point. I will delve deeper into these ambiguities in the Questions section.
>
>
> ### W2. The established theoretical framework predominantly caters to kernel-based dataset distillation methods, and its application seems restricted primarily to the initially proposed KIP method. Given the complexities associated with computing the NTK, KIP's practicality has been empirically questioned. While subsequent kernel-based dataset distillation methods capable of distilling comprehensive datasets have emerged (e.g., [1]), this paper falls short of validating their compatibility with the introduced framework. This oversight not only raises concerns about the paper's soundness but also limits the practicability of the introduced attack method. ([1] Yongchao Zhou, Ehsan Nezhadarya, Jimmy Ba: Dataset Distillation using Neural Feature Regression. NeurIPS 2022)
>
> We understand KIP/NTK has known computational drawbacks. However,
> 1. Through KIP/NTK, we can theoretically understand the effect of data distillation and backdoor. Our new findings can measure the performance of each trigger with a closed-form formula and induce two theory-driven triggers which can survive from the dataset distillation. Advanced kernel based methods lose theoretical clarity.
>
> 2. Practically, we find that our findings on this theoretical framework extend to more advanced approaches, such as FRePo (Dataset Distillation using Neural Feature Regression. NeurIPS, 2022) and DM (Dataset condensation with distribution matching. WACV, 2023). The experimental results show that our triggers can successfully transfer to the FRePo and DM.
>
> | Trigger-type | Dataset | IPC (Image Per Class)} | Distillation | Model | CTA (%) | ASR (%) |
> | ------------ | ------- | ---------------------- | ------------ | ----- | ------- | ------- |
> | Cifar10 | simple-trigger | 10 | FRePO | ConvNet | 60.32 | 83.10 |
> | Cifar10 | relax-trigger | 50 | FRePO | ConvNet | 68.34 | 81.61 |
>
>
> | Trigger-type | Dataset | IPC (Image Per Class)} | Distillation | Model | CTA (%) | ASR (%) |
> | ------------ | ------- | ---------------------- | ------------ | ----- | ------- | ------- |
> | Cifar10 | simple-trigger | 10 | DM |MLP | 36.41 | 77.03 |
> | Cifar10 | simple-trigger | 50 | DM |MLP | 36.88 | 76.79 |
> | Cifar10 | relax-trigger | 10 | DM |MLP  | 36.31 | 76.04 |
> | Cifar10 | relax-trigger | 50 | DM |MLP  | 36.81 | 76.21 |
>
> Overall, we believe we made two major contributions on extending the frontier of backdoor analysis for dataset distillation,
> 1. Deep theoretical understanding.
> 2. The conclusion hold empirically on sota dataset distillation, such as FRePo and DM.

---

> > ### Author Response · Authors · 2023-11-21
> >
> > ### W3. The experimental design in the paper appears insufficient, raising questions about the broader applicability of the proposed attack. First, it relies on a mere two benchmark datasets. Second, the distilled dataset's size variation is limited to IPC (abbreviation of Image Per Class) scenarios of 10 and 50. Third, while cross-architecture generalization is pivotal in dataset distillation, the paper's evaluations seem to be confined to a 3-layer ConvNet, which is consistent with the architecture of the proxy model designated for distillation.
> >
> > We perform a series of experiments corresponding to these concerns:
> > 1. ImageNet:
> > We perform our KIP-based backdoor attack on ImageNet. In our experiment, we randomly choose ten sub-classes to perform our experiment. We also resize each image in the ImageNet into 128x128. The experimental results show that our KIP-based backdoor attack is effective.
> >
> > | Trigger-type   | Dataset  |Model |IPC (Image Per Class) | CTA (%) | ASR (%) |
> > | -------------- | -------- |--- |--------------------- | ------- | --- |
> > | simple-trigger | ImageNet |NTK|10                    | 15.00   | 100.00    |
> > | simple-trigger | ImageNet |NTK|50                    | 16.60   | 100.00   |
> > | relax-trigger  | ImageNet |NTK|10                    | 16.40   | 100.00   |
> > | relax-trigger  | ImageNet |NTK|50                    | 17.00    | 100.00  |
> > 2. IPC:
> >
> > |  Dataset  | Trigger-type | IPC (Image Per Class) | CTA (%)      | ASR (%)  |
> > | -------------- | ------- | --------------------- | ------------ | ------------ |
> > | CIFAR-10 | simple-trigger | 10                    | 41.70 (0.25) | 100 (0.00)   |
> > | CIFAR-10 | simple-trigger | 20                    | 42.58 (0.23) | 100 (0.00)   |
> > | CIFAR-10 | simple-trigger | 30                    | 43.29 (0.35) | 100 (0.00)   |
> > | CIFAR-10 | simple-trigger | 40                    | 43.55 (0.42) | 100 (0.00)   |
> > | CIFAR-10 | simple-trigger | 50                    | 43.66 (0.40) | 100 (0.00)   |
> > |                |         |                       |              |              |
> > | CIFAR-10 | relax-trigger | 10                    | 41.66 (0.01) | 100 (0.00)   |
> > | CIFAR-10 | relax-trigger | 20                    | 42.46 (0.01) | 100 (0.00)   |
> > | CIFAR-10 | relax-trigger | 30                    | 42.99 (0.08) | 100 (0.00)   |
> > | CIFAR-10 | relax-trigger | 40                    | 43.10 (0.09) | 100 (0.00)   |
> > | CIFAR-10 | relax-trigger | 50                   | 43.64 (0.40) | 100 (0.00)   |
> > |                |         |                       |              |              |
> > | GTSRB   | simple-trigger | 10                    | 67.56 (0.60) | 100 (0.00)   |
> > | GTSRB   | simple-trigger | 20                    | 69.44 (0.35) | 100 (0.00)   |
> > | GTSRB   | simple-trigger | 30                    | 70.24 (0.38) | 100 (0.00)   |
> > | GTSRB   | simple-trigger  | 40                    | 70.84 (0.32) | 100 (0.00)   |
> > | GTSRB   | simple-trigger | 50                   | 71.27 (0.24) | 100 (0.00)   |
> > |                |         |                       |              |              |
> > | GTSRB   | relax-trigger   | 10                    | 68.73 (0.67) | 95.26 (0.54) |
> > | GTSRB   | relax-trigger   | 20                    | 70.38 (0.03) | 94.85 (0.13) |
> > | GTSRB   | relax-trigger  | 30                    | 71.26 (0.02) | 95.73 (0.32) |
> > | GTSRB   | relax-trigger | 40                    | 71.81 (0.01) | 95.84 (0.18) |
> > | GTSRB   | relax-trigger | 50                    | 71.54 (0.33) | 95.08 (0.33) |
> >
> >
> > | Dataset | Trigger-type | IPC (Image Per Class)| CTA (%)              | ASR   (%)           |
> > | ------------ | ------- | --- | -------------------- | ----------------- |
> > | CIFAR-10 | Doorping | 10  | 36.35 (0.42) | 80.00 (40.00) |
> > | CIFAR-10 | Doorping | 20  | 37.65 (0.42) | 70.00 (45.83) |
> > | CIFAR-10 | Doorping | 30  | 38.48 (0.36) | 90.00 (30.00)|
> > | CIFAR-10 | DoorPing | 40 | 37.78 (0.61) | 70.00 (45.83)|
> > | GTSRB | DoorPing | 10 | 68.03 (0.92) | 90.00 (30.00)|
> > | GTSRB | DoorPing | 20 | 81.45 (0.46) | 80.00 (40.00)|
> > | GTSRB | DoorPing | 30 | 81.62 (0.71) | 100.00 (0.00)|

---

> > > ### Author Response · Authors · 2023-11-21
> > >
> > > 3. Cross model abilities:
> > > The experiment for cross model ability is presented as follows. We train the distilled dataset poinsoned by simple-trigger and relax-trigger on 3-layers MLP and 3-layers ConvNet. The experimental results show that both CTA and ASR go up as we increase the IPC (Image Per Class), which suggests that the cross model issue may be relieved as the IPC is large enough.
> > >
> > > | Dataset   | Trigger-type | IPC (Image Per Class) | Cross_model | CTA (%)      | ASR(%)    |
> > > | -------------- | ------- | --- | ----------- | ------------ | ------------|
> > > | CIFAR-10| simple-trigger  | 10  | MLP         | 11.58 (2.10) | 40.00 (48.98)  |
> > > | CIFAR-10| simple-trigger  | 10  | CNN         | 47.37 (7.44) | 40.00 (48.98) |
> > > | CIFAR-10| simple-trigger  | 10  | NTK (baseline)        | 41.70 (0.25)  | 100.00 (0.00)   |
> > > |                |         |     |             |              |             |
> > > | CIFAR-10| simple-trigger  | 50  | MLP         | 48.08 (4.72)  | 40.00 (48.98)    |     |
> > > | CIFAR-10| simple-trigger | 50  | CNN         | 95.96 (1.10)  | 100.00 (0.00) |
> > > | CIFAR-10| simple-trigger | 50  | NTK (baseline)         | 43.66 (0.40)  | 100.00 (0.00)   |
> > > |                |         |     |             |              |             |
> > > | CIFAR-10| relax-trigger | 10  | MLP         | 10.52 (7.44)  | 19.40 (38.80) |
> > > | CIFAR-10| relax-trigger | 10  | CNN         | 64.21 (6.98)  | 81.80 (11.44) |
> > > | CIFAR-10 | relax-trigger | 10  | NTK (baseline)         | 41.66 (0.74)  | 100.00 (0.00) |
> > > |                |         |     |             |              |             |
> > > | CIFAR-10 | relax-trigger | 50  | MLP         | 44.24 (4.49)  | 78.28 (24.57)|
> > > | CIFAR-10| relax-trigger | 50  | CNN         | 93.13 (2.24)  | 82.80 (6.53)  |
> > > | CIFAR-10| relax-trigger | 50  | NTK (baseline)         | 43.64 (0.40)  | 100.00 (0.00) |
> > >
> > > ## Question
> > > ### Q1. Regarding Equation 9, why is the objective of the KIP-based backdoor attack to minimize the empirical loss of $f _{\mathcal{S}}$ on either $\mathcal{D}_A$ or $\mathcal{D}_B$, rather than simultaneously reducing the empirical loss on both $\mathcal{D}_A$ and $\mathcal{D}_B$ as your statement about "Backdoor Attack" (The next to the last paragraph above Equation 7)? While you attempt to address this in Equation 10 by introducing to $\tilde{D} = D_A \cup D_B$ establish an upper bound on the loss of $f _{\mathcal{S}}$ with respect to $D$, this formulation seems somewhat unreasonable to me.
> > >
> > > The objective of the KIP-based backdoor attack is to minimize the projection loss and conflict loss, which is based on Equation 10. Although $D$ can be $D_A$ or $D_B$, the inequality (Equation (10) in the paper) holds for both $D=D_A$ and $D=D_B$. Hence, if we can ensure the projection loss and conflict loss are sufficiently low, we can simultaneously ensure the empirical loss (empirical risk) of $\mathcal{D}_A$ and the empirical loss (empirical risk) of $\mathcal{D}_B$ are also sufficiently low. In Equation (10), we introduce $\tilde{D} = D_A \cup D_B$ in order to derive the common upper bound. Assume $\ell(f, (x, y)) = \|f(x)-y\|^2_2$. Equation (10) can be derived by following process,
> > >
> > > $$
> > > \mathbb{E} _{(x, y) \sim D} \ell(f _{\mathcal{S}^*}, (x, y))= \frac{1}{N} \sum _{(x, y) \sim D} \ell(f _{\mathcal{S}^*}, (x, y))
> > > $$
> > >
> > > $$
> > > \leq\frac{1}{N} \sum _{(x, y) \sim \tilde{D}} \ell(f _{\mathcal{S}^*}, (x, y))
> > > $$
> > >
> > > $$
> > >  = \frac{N_A + N_B}{N}[\frac{1}{N_A + N_B}\sum _{(x, y)\sim \tilde{D}} \ell(f _{\mathcal{S}^*}, (x, y))]
> > > $$
> > >
> > > $$
> > > =\frac{N_A + N_B}{N} [\min _{\mathcal{S}} \mathbb{E } _{(x,y)\sim\tilde{D}} \ell(f _{\mathcal{S}}, (x, y))]
> > > $$
> > >
> > > $$
> > >  =\frac{N_A + N_B}{N}
> > > [\min _{\mathcal{S}} \mathbb{E} _{(x,y) \sim \tilde{D}}
> > > \|f _{\mathcal{S}}(x) - y\|_2^2]
> > > $$
> > >
> > > $$
> > > \leq
> > > \frac{N_A + N_B}{N} [\min _{\mathcal{S}}
> > >  \mathbb{E} _{(x,y)\sim\tilde{D}}
> > >  \|f _{\mathcal{S}}(x) - f _{\tilde{D}}(x) + f _{\tilde{D}}(x) - y\| _2^2]
> > > $$
> > >
> > > $$
> > > \leq\frac{N_A + N_B}{N}
> > > [\min _{\mathcal{S}}
> > > \{\mathbb{E} _{(x,y)\sim\tilde{D}} \|f _{\mathcal{S}}(x) - f _{\tilde{D}}(x)\|_2^2
> > > +
> > > \mathbb{E} _{(x,y)\sim\tilde{D}}
> > >  \| f _{\tilde{D}}(x) - y\|_2^2\}]
> > > $$
> > >
> > > Notice that the above inequality holds for both $D = D_A$ and $D=D_B$. Hence, the right hand side of Equation 10 is a common upper bound for empirical risk of $\mathcal{D}_A$ ($\mathbb{E} _{(x, y)\sim D_A} \ell(f _{\mathcal{S}^*}, (x, y))$) and empirical risk of $\mathcal{D}_B$ ($\mathbb{E} _{(x, y)\sim D_B} \ell(f _{\mathcal{S}^*}, (x, y))$).

---

> > > > ### Author Response · Authors · 2023-11-22
> > > >
> > > > ### Q2. It appears that the introduced projection loss can be directly optimized with respect to the trigger $T$. What's the rationale behind setting an upper bound and optimizing the projection loss through this bound? Does this approach offer computational benefits?
> > > >
> > > > Recall that the projection loss is defined as
> > > >
> > > > $$
> > > > \min _{\mathcal{S}} \mathbb{E} _{(x, y) \sim \tilde{D}} \ell(f _{\mathcal{S}}, (x, f _{\tilde{D}}(x)))
> > > > $$
> > > >
> > > > which measures the information loss when we compress the information from larger dataset $\tilde{D}=D_A \cup D_B$ to smaller dataset $D _{\mathcal{S}}$. In our proposed method (relax-trigger), we constrain that $\mathcal{S} = \mathcal{S}_A$. We can get
> > > >
> > > > $$
> > > > \min _{\mathcal{S}} \mathbb{E} _{(x, y)\sim\tilde{D}}
> > > > \ell(f _{\mathcal{S}}, (x, f _{\tilde{D}}(x)))
> > > > \leq\mathbb{E} _{(x, y) \sim \tilde{D}}\ell(f _{\mathcal{S}_A}, (x, f _{\tilde{D}}(x))).
> > > > $$
> > > >
> > > > However, we think that directly optimize the right hand side of above inequality has some drawbacks. The optimal trigger T can induce a model $f _{\tilde{D}}$ such that $f _{\tilde{D}}(x) \approx f _{\mathcal{S}_A}(x)$ for all $(x, y) \in \tilde{D}$. Nevertheless, the $\|f _{\tilde{D}} - f _{\mathcal{S}_A}\| _{\mathcal{H} _k}$ may still very large. In other words, $f _{\tilde{D}}$ may behave very differently with $f _{\mathcal{S} _A}$ when the input is out of the distribution of $\tilde{D}$. Hence, we think that such a dataset $\tilde{D}$ may compress very limited information into $\mathcal{S}_A$. To deal with this issue, we introduce the projection lemma to derive the upper bound of projection loss. By the projection lemma, we first perform orthogonal decomposition on $f _{\tilde{D}}$ as
> > > >
> > > > $$
> > > > f _{\tilde{D}} = f _{1} + f _{2},
> > > > $$
> > > >
> > > > where $f _{1}\in\mathcal{H} _{\mathcal{S} _A} = span [\{ k( \cdot, x) | (x, y) \in \mathcal{S}_A \} ]$ and  $f _{2} \in \mathcal{H} _{\mathcal{S} _A}^{\perp}$. $f_1$ is considered as the model that which can be well-expressed by $\mathcal{S} _A$, while $f _2$ is considered as the model irrelevant with $\mathcal{S} _A$. Hence, we utilize the $f _2$ to construct the upper bound of projection loss. We hope that $f _2$ will tends to $0$ as we optimize the trigger $T$. We believe that the trigger $T$ derived by optimizing our upper bound of projection loss can make $f _{\tilde{D}}$ well compressed into $\mathcal{S} _A$. This is the rationale behind the upper bound of projection lemma.
> > > >
> > > > As for the computational benefits, we do not observe apparent computational benefits or drawbacks when we utilize the upper bound of projection loss to derive the trigger. However, we believe that our theory-driven approach can induce a better trigger as we mentioned above.

---

> > > > > ### Author Response · Authors · 2023-11-22
> > > > >
> > > > > ### Q3. Based on my W3, could you share additional experimental evidence to validate the efficacy of your proposed triggers when applied to models with alternative architectures trained on the synthesized datasets?
> > > > >
> > > > > In our intuition, as we enlarge the size of synthetic dataset, the cross model ability should be relieved. Besides, some existing works (Neural tangent generalization attacks. ICML, 2021) have shown that NTK possesses certain cross model ability. So, we think that the issue of cross model ability can be controlled by tuning the size of the synthetic dataset and choosing suitable NTK. The rigorous behavior of cross model ability related to the size of synthetic dataset is expected to be discussed in the future. The experimental results show that both CTA and ASR go up as we increase the IPC (Image Per Class), which suggests that the cross model issue may be relieved as the IPC is large enough.
> > > > >
> > > > > The experiment for cross model ability is presented as follows. We train the distilled dataset poinsoned by simple-trigger and relax-trigger on 3-layers MLP and 3-layers ConvNet.
> > > > >
> > > > > | Dataset   | Trigger-type | IPC (Image Per Class) | Cross_model | CTA (%)      | ASR(%)    |
> > > > > | -------------- | ------- | --- | ----------- | ------------ | ------------|
> > > > > | CIFAR-10| simple-trigger  | 10  | MLP         | 11.58 (2.10) | 40.00 (48.98)  |
> > > > > | CIFAR-10| simple-trigger  | 10  | CNN         | 47.37 (7.44) | 40.00 (48.98) |
> > > > > | CIFAR-10| simple-trigger  | 10  | NTK (baseline)        | 41.70 (0.25)  | 100.00 (0.00)   |
> > > > > |                |         |     |             |              |             |
> > > > > | CIFAR-10| simple-trigger  | 50  | MLP         | 48.08 (4.72)  | 40.00 (48.98)    |     |
> > > > > | CIFAR-10| simple-trigger | 50  | CNN         | 95.96 (1.10)  | 100.00 (0.00) |
> > > > > | CIFAR-10| simple-trigger | 50  | NTK (baseline)         | 43.66 (0.40)  | 100.00 (0.00)   |
> > > > > |                |         |     |             |              |             |
> > > > > | CIFAR-10| relax-trigger | 10  | MLP         | 10.52 (7.44)  | 19.40 (38.80) |
> > > > > | CIFAR-10| relax-trigger | 10  | CNN         | 64.21 (6.98)  | 81.80 (11.44) |
> > > > > | CIFAR-10 | relax-trigger | 10  | NTK (baseline)         | 41.66 (0.74)  | 100.00 (0.00) |
> > > > > |                |         |     |             |              |             |
> > > > > | CIFAR-10 | relax-trigger | 50  | MLP         | 44.24 (4.49)  | 78.28 (24.57)|
> > > > > | CIFAR-10| relax-trigger | 50  | CNN         | 93.13 (2.24)  | 82.80 (6.53)  |
> > > > > | CIFAR-10| relax-trigger | 50  | NTK (baseline)         | 43.64 (0.40)  | 100.00 (0.00) |

---

> > > > > > ### Comment · Reviewer_bXcG · 2023-11-22
> > > > > >
> > > > > > Thank you for your detailed responses. However, I **maintain my current rating** due to the following reasons:
> > > > > >
> > > > > > 1. In addressing my Weakness 1, your response merely repeats what's described in your paper about Equation 10. However, my primary concern is that **establishing an empirical risk upper bound for a single dataset by using a contrived collection including it and another dataset seems inherently unreasonable**.
> > > > > >
> > > > > > 2. Regarding your answer to my **question 2**, you state that **utilizing the upper bound of projection loss to derive the trigger doesn't offer any additional computational benefits or drawbacks (in the last paragraph)**. However, you appear to argue that **optimizing the upper bound** of your proposed loss **is more advantageous than optimizing the loss itself**. **This argument seems perplexing to me**. Therefore, I recommend elaborating on this part more clearly and adding additional experiments to enhance credibility in your future print.
> > > > > >
> > > > > > 3. It would be recommended to re-organize beneficial supplementary experiments or applications more explicitly in the main text (at least in your paper appendix) of your future print. I believe this could strengthen your argument and the overall presentation of your work.

---

> > > > > > > ### Author Response · Authors · 2023-11-23
> > > > > > >
> > > > > > > ### R1. In addressing my Weakness 1, your response merely repeats what's described in your paper about Equation 10. However, my primary concern is that **establishing an empirical risk upper bound for a single dataset by using a contrived collection including it and another dataset seems inherently unreasonable**.
> > > > > > >
> > > > > > > We would like to note that the introduction of "two datasets" in our formulation is in fact a natural consequence the considered data distillation and backdoor setup, in order to characterize the dependency between the original dataset ($D  _A$) and the malicious dataset ($D _B$).
> > > > > > > Given $D _A$, $D _B$ and denote that $\tilde{D} = D _A \cup D _B$. We introduce two datasets $D _A$ and $D _B$ to bound the empirical loss $\mathbb{E} _{(x, y) \sim D} \ell(f _{\mathcal{S}^*}, (x, y))$ because of the formulation of $f _{\mathcal{S}^*}$. Here, the distilled dataset $\mathcal{S}^*$ is defined as
> > > > > > >
> > > > > > > $$
> > > > > > > \mathcal{S}^* = \arg \min _{\mathcal{S}} \frac{1}{N_A + N_B} \sum  _{(x, y)\in \tilde{D}} \|f _\mathcal{S}(x) - y\|^2_2
> > > > > > > $$
> > > > > > >
> > > > > > > and $f _{\mathcal{S}^*}$ is the model trained on $\mathcal{S}^*$. Hence, the model $f _{\mathcal{S}^*}$ is influenced by $\tilde{D} = D _A \cup D _B$. The empirical loss $\mathbb{E} _{(x, y) \sim D} \ell(f _{\mathcal{S}^*}, (x, y))$ must be simultaneously impacted by $D _A$ and $D _B$. Thus, we must need to use both $D _A$ and $D _B$ to represent the empirical risk $\mathbb{E} _{(x, y) \sim D} \ell(f _{\mathcal{S}^*}, (x, y))$.

---

> > > > > > > > ### Author Response · Authors · 2023-11-23
> > > > > > > >
> > > > > > > > ### R2. Regarding your answer to my **question 2**, you state that **utilizing the upper bound of projection loss to derive the trigger doesn't offer any additional computational benefits or drawbacks (in the last paragraph)**. However, you appear to argue that **optimizing the upper bound** of your proposed loss **is more advantageous than optimizing the loss itself**. **This argument seems perplexing to me**. Therefore, I recommend elaborating on this part more clearly and adding additional experiments to enhance credibility in your future print.
> > > > > > > >
> > > > > > > > This is an insightful comment! In our previous response, our "computational benefits' ' means the consumption of computational time, CTA and ASR. The computation time of optimizing projection loss and the computation time of optimizing upper bound of projection loss do not have an apparent difference. For CTA and ASR, the trigger induced by optimizing the upper bound of projection loss and the trigger induced by optimizing the projection loss show almost equal CTA and ASR. the corresponding experimental results are shown below.
> > > > > > > >
> > > > > > > > | Dataset  |Model| Loss type   | IPC (Image Per Class) | Trans -parency (m) | CTA (%) |ASR (%)| Time (s) |
> > > > > > > > | -------- | ---|-------- | --- | --------------- | ------ | --- | ------ |
> > > > > > > > | CIFAR-10 |NTK |Upperbound | 10  | 0.3             | 41.47 (0.08) | 1.00 (0.00)  | 174.90 (0.63) |
> > > > > > > > | CIFAR-10 | NTK |Project Loss | 10  | 0.3           | 41.46 (0.07) | 1.00 (0.00)   | 173.62 (1.73) |
> > > > > > > > | CIFAR-10 | NTK |Upperbound | 50  | 0.3             | 43.64 (0.28) | 1.00 (0.00)   | 320.98 (0.57) |
> > > > > > > > | CIFAR-10 | NTK |Project Loss | 50  | 0.3           | 43.62 (0.17) | 1.00 (0.00)   | 319.15 (1.34) |
> > > > > > > >
> > > > > > > > As for the advantages of our upper bound mentioned in our previous response, we mean that the model $f _{\tilde{D}}$ induced by trigger $T$ can be better compressed into $\mathcal{S} _A$ in the sense of $\|\cdot\| _{\mathcal{H}_k}$, where $T$ is the trigger derived by optimizing the upper bound. In other words, we believe that $f _{\tilde{D}}(x) \approx f _{\mathcal{S} _A}(x)$ not only for $(x ,y) \in \tilde{D}$ but also for some $(x ,y)$ which are out of the distribution $\tilde{D}$. $f _{\tilde{D}}$ will be similar with $f _{\mathcal{S} _A}$ in a wider domain. That is why we say the trigger derived by optimizing upper bound is better.
> > > > > > > >
> > > > > > > >
> > > > > > > > The value of our upper bound is that it provides us with some unique theoretical insights that are otherwise oblivious in the original loss. The upper bound of projection loss introduces $\|\cdot\| _{\mathcal{H} _k}$ and derives another form to represent the projection loss. We can know that the term
> > > > > > > >
> > > > > > > > $$
> > > > > > > > \|k(\cdot, x_i) - k(\cdot, X _\mathcal{S})
> > > > > > > > k(X _\mathcal{S}, X _\mathcal{S})^{-1} k(X _\mathcal{S}, x_i)\|^2_2
> > > > > > > > $$
> > > > > > > >
> > > > > > > > plays a crucial role when we compress the information of a model $k(\cdot, x_i)$ into the dataset $\mathcal{S}$ because the above formula is small if and only if the upper bound of projection loss is small. As a new theoretical insight, our result show that one can use this term to characterize the projection loss.
> > > > > > > >
> > > > > > > > We will include these discussions in the future version to clarify these points.
> > > > > > > >
> > > > > > > >
> > > > > > > > ### R3. It would be recommended to re-organize beneficial supplementary experiments or applications more explicitly in the main text (at least in your paper appendix) of your future print. I believe this could strengthen your argument and the overall presentation of your work.
> > > > > > > >
> > > > > > > > We will follow the suggestion.

---

### Official Review · Reviewer_Z9NC · 2023-10-30

**Soundness:** 3 good
**Presentation:** 2 fair
**Contribution:** 3 good
**Rating:** 6
**Confidence:** 4

**Summary:**

The research provides a comprehensive exploration of the theoretical underpinnings of backdoor attacks and their interplay with dataset distillation, employing kernel methods as the foundational framework. This investigation leads to the introduction of two innovative trigger pattern generation techniques, intricately crafted to suit the specific requirements of dataset distillation. These methodologies are meticulously derived from a foundation of theoretical insights.

**Strengths:**

1. The research significantly contributes to the theoretical understanding of backdoor attacks and their interaction with dataset distillation. By using kernel methods as the foundational framework, it provides a rigorous theoretical foundation for subsequent developments.

2. The introduction of two novel trigger pattern generation methods tailored for dataset distillation is a notable contribution. These methods are based on theoretical insights and offer new avenues for designing backdoor attacks in this context.

3.The study backs its theoretical findings with comprehensive empirical experiments. The results demonstrate the resilience of datasets poisoned by the designed triggers against conventional backdoor attack detection and mitigation methods, adding practical significance to the research.

**Weaknesses:**

The experimental results presented in the study may benefit from further substantiation to conclusively support the stated claims. A notable observation in Table 2 is that the performance of the 'simple-trigger' method is notably outperformed by 'DoorPing,' which prompts questions regarding the efficacy of the former.

Moreover, enhancing the organization and writing style of the manuscript could enhance its overall readability and comprehension for a wider readership.

**Questions:**

Please refer to "Weaknesses" part.

---

> ### Author Response · Authors · 2023-11-21
>
> # Reviewer Z9NC
> ## Question
> ### Q1. The experimental results presented in the study may benefit from further substantiation to conclusively support the stated claims. A notable observation in Table 2 is that the performance of the 'simple-trigger' method is notably outperformed by 'DoorPing,' which prompts questions regarding the efficacy of the former.
>
> >We perform the following experiments to demonstrate that our proposed method is valid.
> 1. ImageNet:
> We perform our KIP-based backdoor attack on ImageNet. In our experiment, we randomly choose ten sub-classes to perform our experiment. We also resize each image in the ImageNet into 128x128. The experimental results show that our KIP-based backdoor attack is effective.
>
> | Trigger-type   | Dataset  |Model |IPC (Image Per Class) | CTA (%) | ASR (%) |
> | -------------- | -------- |--- |--------------------- | ------- | --- |
> | simple-trigger | ImageNet |NTK|10                    | 15.00   | 100.00    |
> | simple-trigger | ImageNet |NTK|50                    | 16.60   | 100.00   |
> | relax-trigger  | ImageNet |NTK|10                    | 16.40   | 100.00   |
> | relax-trigger  | ImageNet |NTK|50                    | 17.00    | 100.00  |
>
> 2. Transferability:
> We test the transferability of simple-trigger and relax-trigger. We first utilize our triggers to poison the datasets. Then, we distill these datasets with different distillation methods, FRePo (Dataset Distillation using Neural Feature Regression. NeurIPS, 2022) and DM (Dataset condensation with distribution matching. WACV, 2023). The experimental results shows that our triggers can successfully transfer to the FrePo and DM.
>
>
>
> | Trigger-type | Dataset | IPC (Image Per Class)} | Distillation | Model | CTA (%) | ASR (%) |
> | ------------ | ------- | ---------------------- | ------------ | ----- | ------- | ------- |
> | Cifar10 | simple-trigger | 10 | FRePO | ConvNet | 60.32 | 83.10 |
> | Cifar10 | relax-trigger | 50 | FRePO | ConvNet | 68.34 | 81.61 |
>
>
> | Trigger-type | Dataset | IPC (Image Per Class)} | Distillation | Model | CTA (%) | ASR (%) |
> | ------------ | ------- | ---------------------- | ------------ | ----- | ------- | ------- |
> | Cifar10 | simple-trigger | 10 | DM |MLP | 36.41 | 77.03 |
> | Cifar10 | simple-trigger | 50 | DM |MLP | 36.88 | 76.79 |
> | Cifar10 | relax-trigger | 10 | DM |MLP  | 36.31 | 76.04 |
> | Cifar10 | relax-trigger | 50 | DM |MLP  | 36.81 | 76.21 |

---

> > ### Author Response · Authors · 2023-11-21
> >
> > 3. Cross model ability:
> > The experiment for cross model ability is presented as follows. We train the distilled dataset poinsoned by simple-trigger and relax-trigger on 3-layers MLP and 3-layers ConvNet. The experimental results show that both CTA and ASR go up as we increase the IPC (Image Per Class), which suggests that the cross model issue may be relieved as the IPC is large enough.
> >
> > | Dataset   | Trigger-type | IPC (Image Per Class) | Cross_model | CTA (%)      | ASR(%)    |
> > | -------------- | ------- | --- | ----------- | ------------ | ------------|
> > | CIFAR-10| simple-trigger  | 10  | MLP         | 11.58 (2.10) | 40.00 (48.98)  |
> > | CIFAR-10| simple-trigger  | 10  | CNN         | 47.37 (7.44) | 40.00 (48.98) |
> > | CIFAR-10| simple-trigger  | 10  | NTK (baseline)        | 41.70 (0.25)  | 100.00 (0.00)   |
> > |                |         |     |             |              |             |
> > | CIFAR-10| simple-trigger  | 50  | MLP         | 48.08 (4.72)  | 40.00 (48.98)    |     |
> > | CIFAR-10| simple-trigger | 50  | CNN         | 95.96 (1.10)  | 100.00 (0.00) |
> > | CIFAR-10| simple-trigger | 50  | NTK (baseline)         | 43.66 (0.40)  | 100.00 (0.00)   |
> > |                |         |     |             |              |             |
> > | CIFAR-10| relax-trigger | 10  | MLP         | 10.52 (7.44)  | 19.40 (38.80) |
> > | CIFAR-10| relax-trigger | 10  | CNN         | 64.21 (6.98)  | 81.80 (11.44) |
> > | CIFAR-10 | relax-trigger | 10  | NTK (baseline)         | 41.66 (0.74)  | 100.00 (0.00) |
> > |                |         |     |             |              |             |
> > | CIFAR-10 | relax-trigger | 50  | MLP         | 44.24 (4.49)  | 78.28 (24.57)|
> > | CIFAR-10| relax-trigger | 50  | CNN         | 93.13 (2.24)  | 82.80 (6.53)  |
> > | CIFAR-10| relax-trigger | 50  | NTK (baseline)         | 43.64 (0.40)  | 100.00 (0.00) |
> >
> >
> > The result that DoorPing outperforms simple-trigger in Table 2 is consistent to our expectation. In that case ($m=0.3$), relax-trigger should be chosen and will outperform DoorPing. The details are described below.
> >
> > As for Table 2, we aim to say that different mask transparencies will have an impact on CTA/ASR. In the experiments in Table 2, we consider transparencies 30\% (m=0.3). The experimental results show that the ASR of simple-trigger cannot achieve 100\% and is worse than DoorPing's algorithm. However, this consequence is consistent to our theoretical analysis. According to our analysis, we can infer that the generalization gap has only a slight drop as we use the whole image-covered trigger (simple-trigger) when $m=0.3$, which contributes to the poor ASR. So, under this circumstance ($m=0.3$), we need to optimize conflict loss, and projection loss to find a stronger trigger (relax-trigger). Table 2 shows that our relax-trigger is apparently stronger than DoorPing and simple-trigger. To sum up,  Table 2 can show that
> >
> > 1. As our theoretical analysis has predicted, simple-trigger is not useful when the transparency is low enough ($m=0.3$).
> >
> > 2. In a more general circumstance ($m=0.3$), we need to consider another theory-induced trigger, relax-trigger. Furthermore, our experimental results show that the relax-trigger is effective

---

> > > ### Author Response · Authors · 2023-11-21
> > >
> > > ### Q2. Moreover, enhancing the organization and writing style of the manuscript could enhance its overall readability and comprehension for a wider readership.
> > > We will use more straightforward examples to explain the concept of some terms, like conflict loss and projection loss.
> > >
> > > 1. For conflict loss, the conflict loss is defined as
> > > $$
> > >  \mathbb{E} _{(x, y)\sim \tilde{D}}\ell(f _{\tilde{D}}, (x, y))\nonumber,
> > > $$
> > >     where $\tilde{D} = D_A \cup D_B$.
> > >     The conflict measures the information conflict between $D_A$ and $D_B$. For example, we consider a dog/cat picture classification problem. In the dataset $D_A$, we label the dog pictures with $0$ and label the cat pictures with $1$. However, in the dataset $D_B$, we label the dog pictures with $1$ and label the cat pictures with $0$. It is clear that the model trained on $\tilde{D}$ must perform terribly on the dataset either $D_A$ or $D_B$. In this case, the information between $D_A$ and $D_B$ have strong conflict and the conflict loss would be large.
> > >
> > > 2. For projection loss, the projection loss is defined as
> > > $$
> > > \min _{\mathcal{S}} \mathbb{E} _{(x, y)\sim
> > > \tilde{D}}\ell(f _{\mathcal{S}}, (x, f _{\tilde{D}}(x)))\nonumber.
> > > $$
> > >     The projection loss reflects the natural information loss when we compress a large dataset into a small dataset. Take writing an abstract for example. If we want to write a 100 word abstract to  describe a 10000 words article, the abstract may suffer some lack of semantics to some degree. Such a phenomena also happens for dataset distillation. When the information of a large dataset is complex enough, the information loss for dataset distillation will be significant; When the information of a large dataset is very simple, it is possible that there is only very limited information loss. We introduce the projection loss defined above to measure this phenomenon.

---

### Official Review · Reviewer_npoK · 2023-10-31

**Soundness:** 3 good
**Presentation:** 2 fair
**Contribution:** 2 fair
**Rating:** 6
**Confidence:** 3

**Summary:**

The paper aims to bridge a gap in the literature by providing a theoretical framework for understanding backdoor attacks on dataset distillation. It introduces two new theory-driven trigger pattern generation methods: simple trigger and relax trigger, specialized for dataset distillation. The paper presents analyses and experiments on two datasets, showing that these triggers are effective at launching resilient backdoor attacks that can significantly weaken conventional detection and mitigation methods.

**Strengths:**

1. The paper is among the first to provide a theoretical framework for understanding backdoor effects on dataset distillation, thus filling a significant gap in the field.
2. The introduction of simple-trigger and relax-trigger is interesting. These triggers are also shown to be effective through empirical testing.

**Weaknesses:**

1. Some sections could benefit from more straightforward explanations to make the paper more accessible to readers not deeply familiar with the subject matter.
2. The datasets evaluated in the paper appear to be limited in scope. Typically, researchers conduct experiments on more comprehensive datasets like ImageNet, or other comparable datasets, to convincingly demonstrate the effectiveness of a proposed attack method.

**Questions:**

1. Are there some potential defense methods during the dataset distillation process to mitigate backdoor attacks?
2. Given the variety of dataset distillation methods available, could the choice of distillation method potentially impact the conclusions drawn about the efficacy of the proposed attack method?

---

> ### Author Response · Authors · 2023-11-21
>
> # Reviewer npoK
> ## Weakness
> ### W1. Some sections could benefit from more straightforward explanations to make the paper more accessible to readers not deeply familiar with the subject matter.
>
> >We will use more straightforward examples to explain the concept of some terms, like conflict loss and projection loss.
>
> 1. For conflict loss, the conflict loss is defined as
> $$
>  \mathbb{E} _{(x, y)\sim\tilde{D}} \ell(f _{\tilde{D}}, (x, y)),
> $$
>     where $\tilde{D} = D_A \cup D_B$.
>     The conflict measures the information conflict between $D_A$ and $D_B$. For example, we consider a dog/cat picture classification problem. In the dataset $D_A$, we label the dog pictures with $0$ and label the cat pictures with $1$. However, in the dataset $D_B$, we label the dog pictures with $1$ and label the cat pictures with $0$. It is clear that the model trained on $\tilde{D}$ must perform terribly on the dataset either $D_A$ or $D_B$. In this case, the information between $D_A$ and $D_B$ have strong conflict and the conflict loss would be large.
>
> 2. For projection loss, the projection loss is defined as
> $$
> \min _{\mathcal{S}} \mathbb{E } _{(x, y)\sim
> \tilde{D}} \ell(f _{\mathcal{S}}, (x, f _{\tilde{D}}(x)))\nonumber.
> $$
>     The projection loss reflects the natural information loss when we compress a large dataset into a small dataset. Take writing an abstract for example. If we want to write a 100 word abstract to  describe a 10000 words article, the abstract may suffer some lack of semantics to some degree. Such a phenomena also happens for dataset distillation. When the information of a large dataset is complex enough, the information loss for dataset distillation will be significant; When the information of a large dataset is very simple, it is possible that there is only very limited information loss. We introduce the projection loss defined above to measure this phenomenon.
>
>
> ### W2. The datasets evaluated in the paper appear to be limited in scope. Typically, researchers conduct experiments on more comprehensive datasets like ImageNet, or other comparable datasets, to convincingly demonstrate the effectiveness of a proposed attack method.
>
> Following the reviewer's suggestion, we additionally perform our KIP-based backdoor attack on ImageNet. The ASR in our experiments can achieve 100%, which suggests our simple-trigger and relax-trigger are both effective on ImageNet. Here, we randomly choose ten sub-classes to perform our experiment. We also resize each image in the ImageNet into 128x128.
>
> | Trigger-type   | Dataset  |Model |IPC (Image Per Class) | CTA (%) | ASR (%) |
> | -------------- | -------- |--- |--------------------- | ------- | --- |
> | simple-trigger | ImageNet |NTK|10                    | 15.00   | 100.00    |
> | simple-trigger | ImageNet |NTK|50                    | 16.60   | 100.00   |
> | relax-trigger  | ImageNet |NTK|10                    | 16.40   | 100.00   |
> | relax-trigger  | ImageNet |NTK|50                    | 17.00    | 100.00  |

---

> > ### Author Response · Authors · 2023-11-21
> >
> > ## Question
> > ### Q1. Are there some potential defense methods during the dataset distillation process to mitigate backdoor attacks?
> > >We think there are two scenarios for KIP-based backdoor attack:
> >
> > 1. **The distillation process is totally controlled by the attacker.**
> > Under this scenario, the victim does not have any chance to perform defense during the dataset distillation.
> >
> > 2. **The distillation process is controlled by the attacker, but the victim can update the dataset during the distillation process.**
> > Under this scenario, the victim may send his/her dataset class by class to the attacker during the distillation process. By this approach, we can turn the dataset distillation process into a time-varying version. A research result (On the Permanence of Backdoors in Evolving Models. arXiv, 2023, https://arxiv.org/abs/2206.04677) has shown that the backdoor can be removed in the time-varying model. We think that this could be a potential defense. Nevertheless, whether we can extend this concept to distillation-based backdoor attack is still an open problem.
> >
> >
> > ### Q2. Given the variety of dataset distillation methods available, could the choice of distillation method potentially impact the conclusions drawn about the efficacy of the proposed attack method?
> >
> > The choice of distillation method can influence the efficacy of our attack method to some degree. However, in our paper, we can find that KIP faithfully obeys the optimization problem of performance-matching dataset distillations. So, KIP should be a good surrogate of all performance-matching distillation methods. Hence, our KIP-based backdoor attack should remain valid and demonstrate similar behaviors for all performance-matching dataset distillations.
> >
> > For other types of dataset distillations, it depends on whether there exists a kernel which can well approximate the framework of other types of distillations. If we can find such a kernel, then KIP can be a surrogate. Nonetheless, how to construct such a kernel is still an open problem. To the best of our knowledge, there is no literature about this research direction that provides kernel approx for other types of distillation.
> >
> >
> > We perform an experiment for this problem. We first utilize our triggers to poison the datasets. Then, we distill these datasets with different distillation methods, FRePo (Dataset Distillation using Neural Feature Regression. NeurIPS, 2022) and DM (Dataset condensation with distribution matching. WACV, 2023). The experimental results show that our triggers can successfully transfer to the FrePo and DM.
> >
> >
> > | Trigger-type | Dataset | IPC (Image Per Class)} | Distillation | Model | CTA (%) | ASR (%) |
> > | ------------ | ------- | ---------------------- | ------------ | ----- | ------- | ------- |
> > | Cifar10 | simple-trigger | 10 | FRePO | ConvNet | 60.32 | 83.10 |
> > | Cifar10 | relax-trigger | 50 | FRePO | ConvNet | 68.34 | 81.61 |
> >
> >
> > | Trigger-type | Dataset | IPC (Image Per Class)} | Distillation | Model | CTA (%) | ASR (%) |
> > | ------------ | ------- | ---------------------- | ------------ | ----- | ------- | ------- |
> > | Cifar10 | simple-trigger | 10 | DM |MLP | 36.41 | 77.03 |
> > | Cifar10 | simple-trigger | 50 | DM |MLP | 36.88 | 76.79 |
> > | Cifar10 | relax-trigger | 10 | DM |MLP  | 36.31 | 76.04 |
> > | Cifar10 | relax-trigger | 50 | DM |MLP  | 36.81 | 76.21 |

---

> > > ### Comment · Reviewer_npoK · 2023-11-22
> > >
> > > Thank author(s) for your response. I think the author solved most of my concerns but I still think a baseline defense method is necessary for evaluating the attack. I will increase my score.

---

### Official Review · Reviewer_hKa2 · 2023-11-01

**Soundness:** 3 good
**Presentation:** 3 good
**Contribution:** 3 good
**Rating:** 6
**Confidence:** 3

**Summary:**

The paper studies the problem of backdoor attacks to evade data distillation, which introduces subtle changes or "triggers" to data to manipulate machine learning models.  It focuses on the theoretical underpinnings of dataset distillation and its implications on backdoor attacks. Based on the theoretical understandings, the authors propose two new theory-induced trigger generation methods: simple-trigger and relax-trigger. Experimental results demonstrate that these triggers, when used in an attack, can successfully evade common backdoor detection techniques.

**Strengths:**

1. One of the primary strengths of this work is the establishment of the first theoretical framework to understand backdoor effects on dataset distillation. This fills a significant gap in the literature, especially when considering the practical implications of such attacks.
2. The paper introduces two new backdoors - simple-trigger and relax-trigger - which are computationally efficient. The relax-trigger, in particular, is more efficient than DoorPing as it doesn't rely on bi-level optimization.
3. Both the simple-trigger and relax-trigger have been demonstrated to challenge or evade eight existing defense mechanisms.

**Weaknesses:**

1. If we utilize the original dataset instead of the distilled data for model training, would the trigger remain effective? It would be better to include such experiments.
2. Can the proposed attacks evade other data distillation techniques (e.g., gradient matching based methods and distribution matching based methods)? It would further strengthen the experimental evaluation by examining the transferability of the proposed attacks.
3. In my understanding, individuals would majorly employ distilled data for training new models in scenarios such as neural architecture search and continual learning. Expanding on the implications of backdoor attacks in these applications would provide greater clarity.

**Questions:**

1. In Equation (9), why the second term is called the generalization gap?

---

> ### Author Response · Authors · 2023-11-21
>
> # Reviewer hKa2
> ## Weakness
>
> ### W1. If we utilize the original dataset instead of the distilled data for model training, would the trigger remain effective? It would be better to include such experiments.
>
> This is a great suggestion! We perform the experiments on CIFAR-10 and GTSRB. We first utilize the simple-trigger and relax-trigger to poison the dataset. Then, we use 3-layers ConvNet to train a model and evaluate corresponding CTA and ASR. The experimental results demonstrate that our triggers simple-trigger and relax-trigger both remain effective.
>
> | Dataset | Trigger-type | Transparency (m) | CTA (%) | ASR (%) |
> | --- |--- |--- |--- |--- |
> | CIFAR-10 | simple-trigger | 1 | 70.02 (0.40) | 100.00 (0.00) |
> | CIFAR-10 | relax-trigger | 0.3 | 70.02 (0.65) | 99.80 (0.04) |
> | CIFAR-10 | simple-trigger | 0.3 | 67.84 (0.36) | 95.50 (1.23) |
> |  |  |  |  |  ||
> | GTSRB | simple-trigger | 1 |  72.47 (3.36)|100.00 (0.00)  |
> | GTSRB | relax-trigger | 0.3 | 75.50 (2.09) | 99.82 (0.09) |
> | GTSRB | simple-trigger | 0.3 |70.21 (3.03) | 99.36 (0.20) |
>
> ### W2. Can the proposed attacks evade other data distillation techniques (e.g., gradient matching based methods and distribution matching based methods)? It would further strengthen the experimental evaluation by examining the transferability of the proposed attacks.
>
> This is a very insightful question! Our proposed attacks can evade other data distillation techniques. In particular, we perform experiments to examine the transferability of our theory-induced triggers. We first use our simple-trigger and relax-trigger to poison the dataset. Then, we distill dataest with a different distillation method, FRePo (Dataset Distillation using Neural Feature Regression. NeurIPS, 2022) and DM (Dataset condensation with distribution matching. WACV, 2023). The experimental results shows that our triggers can successfully transfer to the FrePo and DM.
>
> The rationale is that KIP faithfully obeys the optimization problem of performance-matching dataset distillations except for our assumption that the models lie in the Reproducing Kernel Hilbert Space (RKHS). So, KIP can be a surrogate for all performance-matching dataset distillations. We believe that our KIP-based backdoor attack should evade all performance-matching dataset distillations, as suggested by our theoretical analysis.  As for other types of dataset distillations, it depends on whether there exists some kernel-like approximation or interpretation to the considered dataset distillation method. If we can find such a kernel-like approximation, then KIP should be a valid surrogate. However, how to find or construct the kernel is still an open question. To the best of our knowledge, there is no literature about this research direction that provides kernel approximation for other types of distillation.
>
> | Trigger-type | Dataset | IPC (Image Per Class)} | Distillation | Model | CTA (%) | ASR (%) |
> | ------------ | ------- | ---------------------- | ------------ | ----- | ------- | ------- |
> | Cifar10 | simple-trigger | 10 | FRePO | ConvNet | 60.32 | 83.10 |
> | Cifar10 | relax-trigger | 50 | FRePO | ConvNet | 68.34 | 81.61 |
>
>
> | Trigger-type | Dataset | IPC (Image Per Class)} | Distillation | Model | CTA (%) | ASR (%) |
> | ------------ | ------- | ---------------------- | ------------ | ----- | ------- | ------- |
> | Cifar10 | simple-trigger | 10 | DM |MLP | 36.41 | 77.03 |
> | Cifar10 | simple-trigger | 50 | DM |MLP | 36.88 | 76.79 |
> | Cifar10 | relax-trigger | 10 | DM |MLP  | 36.31 | 76.04 |
> | Cifar10 | relax-trigger | 50 | DM |MLP  | 36.81 | 76.21 |

---

> > ### Author Response · Authors · 2023-11-21
> >
> > ### W3. In my understanding, individuals would majorly employ distilled data for training new models in scenarios such as neural architecture search and continual learning. Expanding on the implications of backdoor attacks in these applications would provide greater clarity.
> >
> > We agree that inclduing the results from neural architecture search and continual learning can further strengthen our contributions. We train our distilled dataset poinsoned by simple-trigger and relax-trigger in different scenarios, neural architecture search (NAS) and continual learning (CL). The experimental results are shown below.
> >
> >
> > The experiemnt result below shows that our triggers remain effective for NAS.
> >
> > | Trigger-type   | Dataset | IPC (Image Per Class) | Scenario | CTA (%)      | ASR (%)      |
> > | -------------- | ------- | --------------------- | -------- | ------------ | ------------ |
> > | simple-trigger | CIFAR-10 | 50                    | NAS      | 37.49(3.44) | 100.00(0.00) |
> > | relax-trigger  | CIFAR-10 | 50                    | NAS      | 36.43(3.62) | 86.23(3.22)  |
> >
> > >For CL, the experiment result belwo shows that both CTA and ASR are slightly higher than baseline.
> >
> > | Trigger-type   | Dataset | IPC (Image Per Class) | Scenario | CTA (%)      | ASR (%)      |
> > | -------------- | ------- | --------------------- | -------- | ------------ | ------------ |
> > | simple-trigger | CIFAR-10 | 50                    | CL       | 13.93(1.93) | 100.00(0.00) |
> > | simple-trigger | CIFAR-10 | 50                    | baseline | 13.60(1.66) | 100.00(0.00) |
> > | relax-trigger  | CIFAR-10 | 50                    | CL       | 20.13(2.94) | 60.94(21.68) |
> > | relax-trigger  | CIFAR-10 | 50                    | baseline | 14.00(3.54)  | 43.11(7.83)  |
> >
> > Note that the details about our implementation of NAS and CL are described below.
> >
> > >NAS: The process defines a search space (random_search) that includes a range of possible model parameters such as the number of convolutional layers, the number of dense layers, and the size of the convolutional layers. The program randomly selects parameters from this space to generate multiple candidate model architectures. A CNN model is then built, comprising convolutional layers (Conv2D), batch normalization (BatchNormalization), activation functions (such as ReLU), pooling layers (MaxPooling2D), flattening layers (Flatten), fully connected layers (Dense), and optionally Dropout layers. Each model is compiled and trained using the Adam optimizer and categorical cross-entropy loss function, but in this case, the same dataset is used for evaluation (although typically, an independent test set should be used). The accuracy and loss functions of different models are compared, and ultimately the best model is selected and saved
> >
> > >CL: The dataset is divided into different category-specific subsets (as in CIFAR-10, which is divided into 10 categories), each containing images and their corresponding labels. This allows the model to gradually train on each subset. A CNN model is built, including multiple convolutional layers (Conv2D), batch normalization layers (BatchNormalization), ReLU activation functions, max pooling layers (MaxPooling2D), and fully connected layers (Dense). The final layer uses a softmax activation function, a typical configuration for label classification tasks. The model is compiled using an RMSprop optimizer and categorical cross-entropy loss function. Further training optimization can be applied, such as using Elastic Weight Consolidation (EWC) to minimize the impact on the originally trained model when learning new subsets.
> >
> > ## Question
> > ### Q1. In Equation (9), why the second term is called the generalization gap?
> >
> > Given a model $f_{\mathcal{S}^*}: \mathcal{X}\rightarrow\mathcal{Y}$, a dataset $D\sim\mathcal{D}^N$, the generalization gap is defined as
> > $$
> >         \mathbb{E} _{(x,y)\sim\mathcal{D}} \ell(f _{\mathcal{S}^*}, (x, y))
> >         - \mathbb{E} _{(x, y) \sim
> >         - D} \ell(f _{\mathcal{S}^*}, (x, y)),
> > $$
> > where $\ell(\cdot, \cdot)$ is some loss function, $\mathbb{E} _{(x, y)\sim\mathcal{D}} \ell(f _{\mathcal{S}^*}, (x, y))$ is called risk and $\mathbb{E} _{(x, y) \sim D} \ell(f _{\mathcal{S}^*}, (x, y))$ is called empirical risk. This mathematical formulation suggests the gap between the performances of $f _{\mathcal{S}^*}$ applied on $D$ and $\mathcal{D}$, respectively. The above equation (Equation (9) in the paper) also indicates whether the performance of $f _{\mathcal{S}^*}$ on $D$ can be generalized to $\mathcal{D}$. Hence, we call the second term in Equation (9) as generalization gap.

---

> > > ### Comment · Reviewer_hKa2 · 2023-12-04
> > > **Thanks for the reponse**
> > >
> > > Thanks for the detailed response. I have no further questions and I will maintain my score.

---

### Meta-Review · Area_Chair_Y416 · 2023-12-05

**Metareview:**

This work proposes a new backdoor attack on dataset distillation motivated by their theoretical work on this subject, specifically for kernel-based dataset distillation approaches.  The reviewers raised several concerns which were mostly addressed by the authors during the discussion period.  This work is a bit niche and may draw a limited audience.  Nonetheless, dataset distillation has been growing in popularity (not necessarily in practice but in research), so I am inclined to accept this paper.

**Justification For Why Not Higher Score:**

The paper is a bit niche.

**Justification For Why Not Lower Score:**

The theoretical work motivates the empirical methodology, which makes empirical improvements to backdoors.  The presentation is clear, and the evaluations are thorough.

---

### Decision · Program_Chairs · 2024-01-16

Accept (poster)